# Engineering viral vectors for acoustically targeted gene delivery

Hongyi R. Li[1], Manwal Harb[2], John E. Heath [1], James S. Trippett[2], Mikhail G. Shapiro [3,4,5] ✉ & Jerzy O. Szablowski [2,3,6,7] ✉

Targeted gene delivery to the brain is a critical tool for neuroscience research and has significant potential to treat human disease. However, the site-specific delivery of common gene vectors such as adeno-associated viruses (AAVs) is typically performed via invasive injections, which limit its applicable scope of research and clinical applications. Alternatively, focused ultrasound blood-brain-barrier opening (FUS-BBBO), performed noninvasively, enables the site-specific entry of AAVs into the brain from systemic circulation. However, when used in conjunction with natural AAV serotypes, this approach has limited transduction efficiency and results in substantial undesirable transduction of peripheral organs. Here, we use high throughput in vivo selection to engineer new AAV vectors specifically designed for local neuronal transduction at the site of FUS-BBBO. The resulting vectors substantially enhance ultrasound-targeted gene delivery and neuronal tropism while reducing peripheral transduction, providing a more than ten-fold improvement in targeting specificity in two tested mouse strains. In addition to enhancing the only known approach to non-invasively target gene delivery to specific brain regions, these results establish the ability of AAV vectors to be evolved for specific physical delivery mechanisms.

Gene therapy is one of the most promising emerging approaches to treating human disease. Recently, a number of gene therapies were approved for clinical use to treat diseases such as blindness[1], muscular dystrophy[2], and metabolic disorders[3] with Adeno-Associated Viral vectors (AAVs). Gene therapy could also potentially target brain disorders. Unfortunately, gene delivery to the brain remains a major challenge. The typical approach for the administration of such gene therapies involves a surgical injection directly into the brain parenchyma, which is invasive. Other studies show it may also be possible to achieve brain-wide gene delivery with systemic[4–6] or intrathecal

injections[7]. However, these approaches, while noninvasive, lack spatial precision and thus cannot target regionally defined neural circuits.

Focused ultrasound blood-brain barrier opening (FUS-BBBO) has the potential to overcome these limitations by providing a route to noninvasive, site-specific gene delivery to the brain[8–12]. In FUS-BBBO ultrasound is focused through an intact skull[13,14] to transiently loosen tight junctions in the BBB and allow for the passage of AAVs from the blood into the targeted brain site. Other mechanisms of FUS-BBBO could include increased transcytosis[15] and decreased levels of efflux transporters[16]. FUS-BBBO can target intravenously administered AAVs

[1]Division of Biology and Biological Engineering, California Institute of Technology, Pasadena, CA, USA. [2]Department of Bioengineering, Rice University, Houston, TX, USA. [3]Division of Chemistry and Chemical Engineering, California Institute of Technology, Pasadena, CA, USA. [4]Andrew and Peggy Cherng Department of Medical Engineering, California Institute of Technology, Pasadena, CA, USA. [5]Howard Hughes Medical Institute, Pasadena, CA, USA. [6]Rice Neuroengineering Initiative, Rice University, Houston, TX, USA. [7]Rice Synthetic Biology Institute, Rice University, Houston, TX, USA. ✉ e-mail: mikhail@caltech.edu; jszab@rice.edu

to millimeter-sized brain sites or cover large regions of the brain without apparent tissue damage in the tested timeframes[17,18]. These capabilities place FUS-BBBO in contrast with intraparenchymal injections, which are invasive and deliver genes to a single 2–3 millimeter-sized region per injection[19,20], requiring a large number of brain penetrations to cover larger regions of interest. At the same time, the spatial targeting capability of FUS-BBBO differentiates it from the use of spontaneously brain-penetrating engineered AAV serotypes which lack spatial specificity[5]. In proof of concept studies, FUS-BBBO has been used in rodents to introduce AAVs encoding reporter genes such as GFP[8,9,17,21], growth factors[22], and optogenetic receptors[10]. The delivery of chemogenetic receptors to the hippocampus provided the ability to modulate memory formation[11].

Despite its promise, three critical drawbacks currently limit the potential of FUS-BBBO in research and therapy applications. First, the BBB effectively limits the transduction of systemically administered naturally occurring AAVs in non-FUS-targeted regions. peripheral organs have endothelia that allow AAV entry and consequently receive a high dose of the virus, which could lead to toxicity[23]. Second, the relative inefficiency of AAV entry at the site of FUS-BBBO have led published studies to use doses that were higher than those needed for direct intraparenchymal injections, which in the clinic typically range from $10^{10}$ to $10^{12}$ viral genomes (VGs) per site injected, compared to $10^{12}$–$10^{14}$ VGs per kilogram of body weight for intravenous route[24]. In our previous work, to achieve transduction efficiency comparable to such injections at $5 \times 10^8$ VGs, we used $10^{10}$ VGs per gram of body weight intravenously with FUS-BBBO[11]. The AAV9 doses used in other FUS-BBBO studies to date have ranged from $5 \times 10^8$ to $1.67 \times 10^{10}$ VGs per gram of body weight[8,9,11,21,25,26]. Lowering the viral doses would reduce the chances of peripheral toxicity, and the costs of potential therapies[24].

We reasoned that these limitations arise from the fact that wild-type serotypes of AAV did not evolve to cross physically loosened biophysical barriers and are therefore not optimal for this purpose. We hypothesized that we could address these limitations by developing new engineered viral serotypes specifically optimized for FUS-BBBO delivery. Capsid engineering techniques[27] in which mutations are introduced into viral capsid proteins have been used to enhance gene delivery properties such as tissue specificity[5,6,28–30], immune evasion[31], or axonal tracing[32]. However, they have not yet been used to optimize viral vectors to work in conjunction with specific physical delivery mechanisms.

To test our hypothesis, we performed in vivo selection of mutagenized AAVs in mice in conjunction with FUS-BBBO (Fig. 1) by adapting a recently developed Cre-recombinase-based screening methodology[6,30]. We identified 5 viral capsid mutants with enhanced transduction at the site of FUS-BBBO and not in the untargeted brain regions. We then performed detailed validation experiments comparing each of these mutants to the parent wild-type AAV, revealing a significant increase in on-target transduction efficiency, increased neuronal tropism, and a marked decrease in off-target transduction in peripheral organs, with an overall performance improvement of more than 10-fold. These results demonstrate the evolvability of AAVs for specific physical delivery methods.

## Results

### High-throughput in vivo screening for AAVs with efficient FUS-BBBO transduction

To identify new AAV variants with improved FUS-BBBO-targeted transduction of neurons, we generated a library of viral capsid sequences containing insertions of 7 randomized amino acids between residues 588 and 589 of the AAV9 capsid protein (Supplementary Fig. S1). Such 7-mer insertions have been widely used to engineer AAVs with new properties[5,6,27–32]. We chose AAV9 as a starting point due to its use in previous FUS-BBBO studies[8,9,11] and superior transduction compared to other naturally occurring AAV serotypes[21].

To make the screening more efficient, we employed recombination-based AAV selection[6,30]. This approach uses a Cre recombinase inside the cells to invert a fragment of the vector's DNA. (Supplementary Fig. S1a). Because Cre is only present inside the cells, this approach allows for the identification of capsid variants that can enter the cells and deliver their DNA to the nucleus. These Cre-inverted DNA sequences can then be detected by PCR using primers specific to the inverted section of the DNA (Supplementary Fig. S1b). Here, we used transgenic mice that expressed Cre in neurons, to select for AAVs with improved neuronal transduction[5,6,33].

To ensure we selected for AAVs transduced specifically within the FUS-BBBO-targeted areas we started with a library of $1.3 \times 10^9$ AAV candidates delivered to one hemisphere with FUS-BBBO (Fig. 1a, b). We then extracted the viral DNA that was delivered to the targeted hemisphere, and re-screened the extracted variants again to quantify specificity and efficiency of FUS-BBBO-mediated transduction. We targeted 4 sites within one hemisphere using magnetic resonance imaging (MRI) guidance, and confirmed the successful BBB opening through imaging of gadolinium contrast agent extravasation (Fig. 2a). We employed FUS parameters below tissue damage limits[11,34] (0.33 MPa at 1.5 MHz, 10 ms pulse length, 1 Hz repetition frequency, 0.22 µl dose of microbubbles per gram of body weight). The AAV libraries were delivered intravenously (IV) immediately following FUS application to the brain at a dose of at a dose of $6.7 \times 10^9$ VGs per gram of body weight. We then allowed for 2 weeks of expression, euthanized the mice for tissue collection. Immediately after, we extracted the viral DNA from the brain and used Cre-dependent PCR amplification to selectively amplify the Cre-modified viral DNA, with a goal of finding

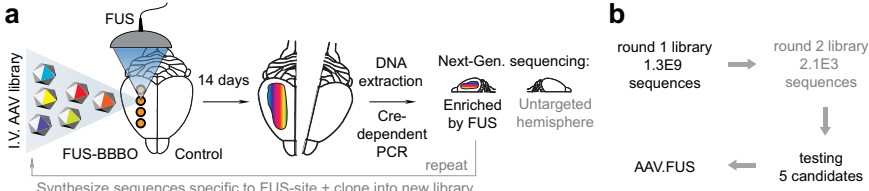

**Fig. 1 | Screening methodology for generation of an AAV for improved site-specific noninvasive gene delivery to the brain. a** Summary of the high-throughput screening and selection process. AAV library is administered intravenously (I.V.) and delivered to one brain hemisphere through FUS-BBBO. After 14 days mice are euthanized, their brain harvested, and the DNA from selected hemispheres is extracted. The DNA is then amplified by Cre-dependent PCR that enriches the viral DNA modified by Cre. In our case, neurons expressed Cre exclusively, and the Cre-dependent PCR enriched viral DNA of AAVs that transduced neurons. We subjected the obtained viral DNA to next-generation sequencing for the targeted hemisphere (round 1) or both targeted and control hemispheres (round 2). The process is then repeated for the next round (steps exclusive to round 2 indicated by the gray text). **b** Overall, 1.3 billion clones were screened in the first round, and 2098 clones in the second round of selection. Out of these clones, we selected 5 that were tested in low-throughput to yield AAV.-FUS.3–a vector with enhanced FUS-BBBO gene delivery.

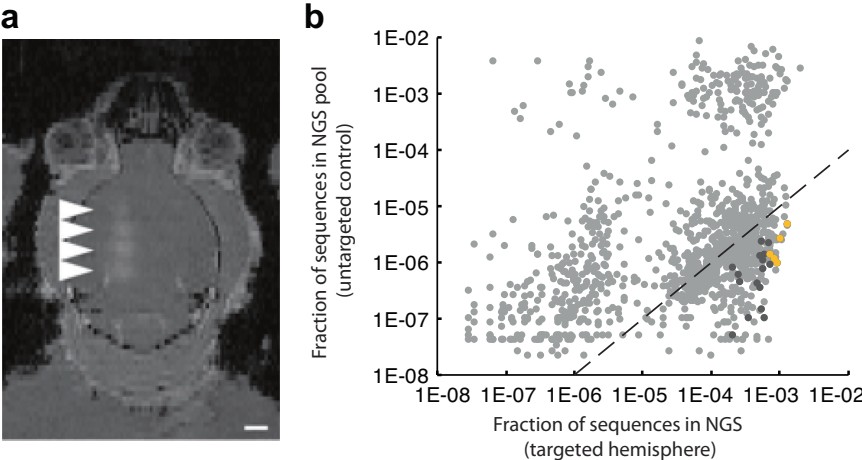

**Fig. 2 | High throughput screening yields vectors with improved FUS-BBBO gene delivery. a** An MRI image showing mouse brain with 4 sites opened with FUS-BBBO in one hemisphere. The bright areas (arrowheads) indicate successful BBB opening and extravasation of the MRI contrast agent Prohance into the brain. This BBB opening was used for delivery of the AAV library. **b** Sequencing results of round 2 of screening show a fraction of NGS reads within the DNA extracted from brains of Syn1-Cre mice subjected to FUS-BBBO and injected with a focused library of 2098 clones. Each dot represents a unique capsid protein sequence, and the position on each axis corresponds to the number of times the sequence was detected in the FUS-targeted and untargeted hemispheres. Markers below the dotted line represent sequences that on average showed 100-fold higher enrichment in the targeted hemisphere as compared to the control hemisphere. Dark gray dots represent 22 clones that are enriched in the FUS targeted hemispheres at least 100-fold in every tested mouse and DNA sequence encoding the 7-mer insertion peptide. Additional 13 clones had zero detected transduction in the untargeted hemisphere and could not be presented on the log-log plot. Yellow dots represent 5 clones (AAV.FUS.1-5) selected for low-throughput testing. Due to the use of a logarithmic plot, clones that had zero copies detected in either of the hemispheres are not shown. Data from one male and one female mouse.

AAVs selectively transducing neurons. We then sequenced the obtained DNA with next-generation sequencing (NGS) of the region of the 7-mer insertion and selected the 2098 most abundant sequences for subsequent evaluation. This screen selected for AAVs which could enter the neurons. However, these variants could not be quantitatively compared at this stage, due to large number of vectors in library compared to the total administered dose. As a result, each AAV clone existed in the library in a small copy number preventing statistically meaningful comparisons between each AAV candidates.

Instead, to quantitatively compare our 2098 down-selected capsid variants, we re-synthesized and packaged them as a new AAV library at a dose of $1.3 \times 10^9$ viral genomes per gram of body weight, corresponding to ~$1.5-3 \times 10^7$ viral genomes of each clone being injected into each mouse. In each of the two hSyn-CRE mice, we injected the AAV library intravenously and opened the BBB in one hemisphere using MRI-guided FUS as in round 1. Two weeks after treatment, we performed a series of procedures on each mouse. First, we removed the brain and separated the two hemispheres. We then extracted DNA from both the hemisphere that was targeted by the FUS and the hemisphere that was not. The DNA extract was amplified by the CRE-dependent PCR to enrich for viral genomes that transduced neurons. After FUS-BBBO delivery, DNA extraction, CRE-dependent PCR, and NGS, we recovered 1433 sequences.

To identify the most improved candidates, we examined their copy number in each hemisphere (Fig. 2b). To identify AAVs that selectively transduced sites that underwent FUS-BBBO, we first looked for variants that were at least 100-fold more represented in the targeted hemisphere relative to the untargeted hemisphere. From this list, we further selected candidates for which the 100-fold difference was maintained in both mice. To ensure that the sequences were not the result of sequencing error, we selected candidates that were found with two alternative codon sequences corresponding to its 7-mer peptide. In the end, 35 sequences met these criteria (dark gray symbols in Fig. 2b). Among these FUS-BBBO-specific variants, we chose the 5 most common sequences, which we hypothesized would code for AAV capsids with the most efficient neuronal transduction. We re-synthesized these sequences

(Supplementary Table 1), cloned them into the AAV9 capsid between amino acids 587–588, and packaged them for detailed evaluation, naming them AAV.FUS 1 through 5.

## AAV.FUS candidates show enhanced transduction of neurons in targeted brain regions and reduced transduction of peripheral organs

An ideal AAV vector for ultrasound-mediated gene delivery to the brain would efficiently transduce targeted neurons while avoiding the transduction of peripheral tissues, such as the liver which is highly transduced by the naturally-occurring AAV serotypes[35]. Additionally, such a vector should only transduce the brain at the FUS-targeted sites. Of the natural AAV serotypes, AAV9 is most commonly used in FUS-BBBO because it transduces neurons at the ultrasound target with relatively high specificity and efficiency compared to untargeted brain regions[8,10,11,21]. However, AAV9 also shows peripheral transduction and is typically administered at doses higher than those used in direct intraparenchymal injection[8,10,11], leaving room for improvement. To evaluate our engineered vectors, we used AAV9 as a benchmark and an internal control for each tested animal.

We performed FUS-BBBO while intravenously co-administering each AAV.FUS candidate alongside AAV9 in individual comparison experiments at 1E10 VGs per gram of body weight. Consequently, each mouse had an internal control where the injected volume, targeted brain site, and the efficiency of FUS-BBBO were identical for both serotypes, leaving the efficiency of the vector as the independent variable. To quantify the transduction efficiency, we encoded the fluorescent proteins mCherry and EGFP in AAV9 and each AAV.FUS variant, respectively, under a cell-type nonselective CaG promoter[36]. After 2 weeks of expression, we counted the numbers of mCherry and EGFP-expressing cells within the sites of FUS-BBBO. We established the reliability of this quantification method by comparing cell counts in the brain for co-administered AAV9-EGFP and AAV9-mCherry (Supplementary Fig. S2). Our quantification showed that AAV.FUS.1, 2, 3, and 5 had significantly improved transduction efficiency compared to AAV9 ($p = 0.0274$, $0.0003$, $0.0052$, $0.0087$, respectively, two-way ANOVA with Sidak's multiple comparisons test, $F_{(4,24)} = 59.49$, Fig. 3a, b)

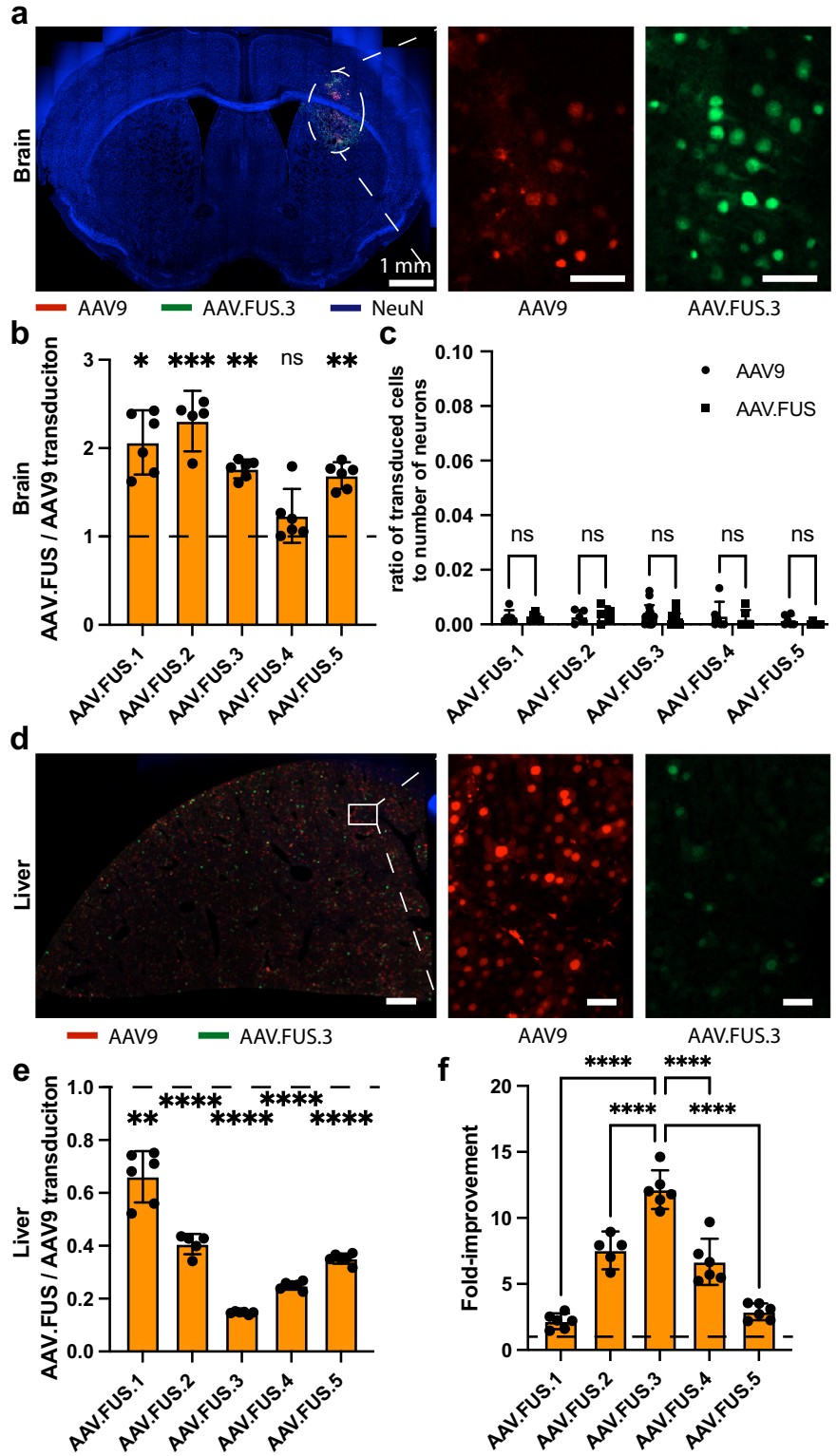

whereas AAV.FUS.4 showed no improvement ($p = 0.2556$). The fold-change in transduction relative to AAV9 was greatest for AAV.FUS.2, and lowest for AAV.FUS.4 (Supplementary Fig. S3). None of the AAV.-FUS candidates produced substantial off-target expression within the brain at sites not insonated by FUS, with AAV9 producing $0.29 \pm 0.1\%$ neuronal transduction ($n = 40$ mice), AAV.FUS.3 $0.17 \pm 0.1\%$ ($n = 17$ mice), and other AAV.FUS candidates between $0.24 \pm 0.12\%$ ($n = 6$), $0.37 \pm 0.26\%$ ($n = 5$), $0.2 \pm 0.26\%$ ($n = 6$), $0.026 \pm 0.05\%$ ($n = 6$) for AAV.FUS.1, 2, 4, and 5 respectively (Fig. 3c).

Next, we evaluated the extent to which AAV.FUS candidates transduce off-target peripheral organs. In mice that received intravenous co-injections of AAV9-mCherry and each variant of AAV.FUS-EGFP, we counted transduced cells in the liver, a peripheral organ known to be targeted by AAVs and a potential source of dose-limiting toxicity[37,38]. Two weeks after injection, we imaged liver sections and counted cells expressing each fluorophore (Fig. 3d, e). We found markedly reduced liver transduction among the AAV.FUS candidates compared to AAV9 (Fig. 3e). AAV.FUS 3 showed the

**Fig. 3 | AAV.FUS candidates improve efficiency of gene delivery to the brain and reduce peripheral transduction. a** Representative images were obtained from mice co-injected with AAV9 and a AAV.FUS.3 at $10^{10}$ viral genomes per gram of body weight each. Sections show brain transduction by AAV9 (red) and AAV.FUS.3 (green), and are counterstained with a neuronal stain (NeuN, blue). **b** All but one (AAV.FUS.4) AAV.FUS candidates showed significant improvement over the co-injected AAV9. (*p*-values for AAV.FUS.1-5, *p* = 0.0274, 0.0003, 0.0052, 0.2556, 0.0087, respectively; Two-way ANOVA with Sidak multiple comparisons test: $F_{(1, 24)}$ = 59.49, P value; *P* < 0.0001). Data from 3 male and 3 female mice per serotype. **c** We found that few cells were transduced outside of the FUS-targeted site and AAV.FUS.3 and AAV9 were not significantly different. (0.19% vs 0.4%, respectively; *p* = 0.072, two-way ANOVA with Sidak multiple comparisons test; $F_{(1, 35)}$ = 2.457, *p* = 0.1260). Similarly, other candidates also showed no differences compared to AAV9 (AAV.FUS.1, *p* = 0.99; AAV.FUS.2, *p* = 0.98; AAV.FUS.4, *p* = 0.86; AAV.FUS.5, *p* = 0.83). Data from 3 male and 3 female mice for all serotypes, except AAV.FUS.2 (2 male, 3 female mice), and AAV.FUS.3 (8 male and 8 female mice). **d** Representative images showing liver transduction by AAV9 (red) and AAV.FUS.3 (green). **e** All tested candidates showed reduced liver transduction as compared to the co-injected AAV9 in the same mice for which brain expression was analyzed. (*P*-values for AAV9 vs AAV.FUS.1-5 were *p* = 0.0058 for AAV.FUS.1, and *p* < 0.0001 for other candidates; Two-way ANOVA, $F_{(1, 24)}$ = 375.9, *P* < 0.0001. Data from 3 male and 3 female mice for all serotypes, except AAV.FUS.2 (2 male, 3 female mice). **f** We defined the fold-improvement in targeting efficiency as the ratio of brain transduction to the liver transduction efficiency using AAV9 as a baseline, which suggested that AAV.FUS.3 is the top candidate for further study. (AAV.FUS.3 compared to AAV.FUS.1,2,4,5, all *p*-values were *p* < 0.0001, one way ANOVA with Tukey HSD post hoc comparison test). Scale bars are 50 μm in (**a, c**), unless otherwise noted. (****p* < 0.0001; ***p* < 0.001; **p* < 0.01; *p* < 0.05, ns = not significant); Error bars are 95% CI. The numbers of animals used in each experiment were: Data from 3 male and 3 female mice per serotype, except AAV.FUS.2 (2 male, 3 female mice). Center for the error bars represents arithmetic mean in (**b, c, e, f**).

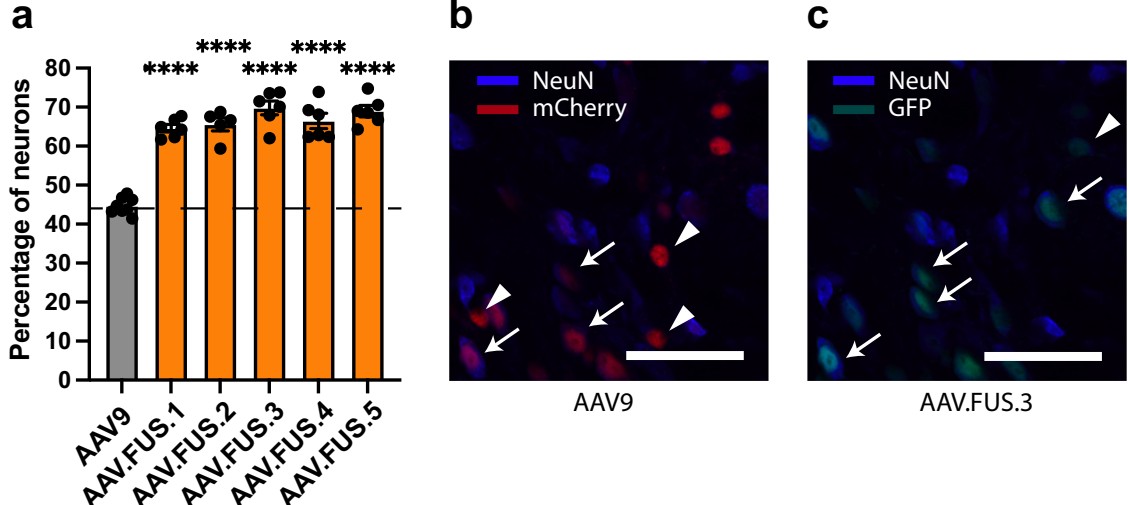

**Fig. 4 | AAV.FUS candidates show improved neuronal tropism. a** All AAV.FUS candidates show improved neuronal tropism upon FUS-BBBO gene delivery. AAV.FUS.3 had 56% more likelihood of transducing a neuron than AAV9 (69.8%, vs 44.7% neuronal transduction, respectively; (for all samples *p* < 0.0001, one way ANOVA, $F_{(5, 31)}$ = 52.60, *P* < 0.0001; *n* = 8 for AAV9, *n* = 6 for all AAV.FUS.1,3,4,5, *n* = 5 for AAV.FUS.2, center for the error bars represents arithmetic mean.). **b** Representative images showing AAV9 transducing both neurons (blue, NeuN staining, example neurons designated by an arrow) and non-neuronal cells (example non-neuronal cells designated by an arrowhead). **c** In comparison, more of the cells transduced with AAV.FUS (green) are neurons (example neurons designated by an arrow), rather than non-neuronal cells (example cell designated by an arrowhead). IV injection dose, $10^{10}$ vg/g. Scale bars are 50 μm. (****p* < 0.0001). Error bars are 95% CI.

largest reduction in liver transduction compared to the wild-type serotype (6.8-fold reduction, *p* < 0.0001, one-way ANOVA with Tukey-HSD post hoc test; $F_{(4, 24)}$ = 93.91), which was significantly higher reduction compared to the other tested AAV.FUS candidates (Supplementary Fig. S4). We did not observe substantial transduction in kidney and lung sections transduced with either viral vector (Supplementary Fig. S5, *n* = 6 mice per organ tested), which is consistent with the published data for the parent AAV9[21]. In kidneys, we observed areas of red autofluorescence, which is consistent with previous reports even in the absence of AAV delivery[39]. This autofluorescent signal did not interfere with detection of AAV.FUS.3, which showed no positive signal in these areas (Fig. S5a).

Our analyses of brain and liver transduction showed that AAV.FUS candidates both decrease the targeting of the liver and increase the transduction efficiency of the targeted brain regions, which leads to a large overall improvement in transduction specificity, expressed as the ratio of the fold-increase in brain transduction and the fold-decrease in liver transduction compared to AAV9. By this metric, AAV.FUS.3 showed a 12.1-fold improvement, significantly greater than the other candidates (*p* < 0.0001 for all pairwise comparisons, one-way ANOVA with Tukey-HSD post hoc test; $F_{(4, 24)}$ = 70.88; Fig. 3f).

Representative images can be found in Supplementary Fig. S6, and detailed sequence data in Appendix A.

A final criterion for successful gene delivery in many applications is the ability to transduce specific cell types at the targeted anatomical location, such as neurons. AAV9 transduces both neuronal and non-neuronal cell types[40–42]. We hypothesized that, since our Cre-dependent screen used mice with the recombinase expressed under a neuronal promoter, our engineered variants could have a higher neuronal tropism relative to their wild-type parent serotype. To test this hypothesis, we immunostained brain sections from mice co-transduced with AAV9-mCherry and each variant of AAV.FUS-EGFP during FUS-BBBO for the neuronal marker NeuN and imaged these sections for GFP, mCherry, and NeuN signal. The fraction of AAV9-transduced (mCherry-positive) cells that were also positive for NeuN was 44.7% (±1.5%, 95% CI; *n* = 8). In contrast, all AAV.FUS candidates had higher neuronal tropism (*p* < 0.0001 for all AAV.FUS candidates, Fig. 4), with neurons constituting between 64.6% (±1.9%, 95% CI; *n* = 6, AAV.FUS.1) and 69.8% (±3.5%, 95% CI, *n* = 6, AAV.FUS.3) of all transduced cells. AAV9 and AAV.FUS transduced astrocytes to a comparable degree (8% vs 3.4% respectively; *n* = 6 sections analyzed from *n* = 3 mice, *p* = 0.0552, paired *t* test; *t* = 4.076). However, AAV9 transduced

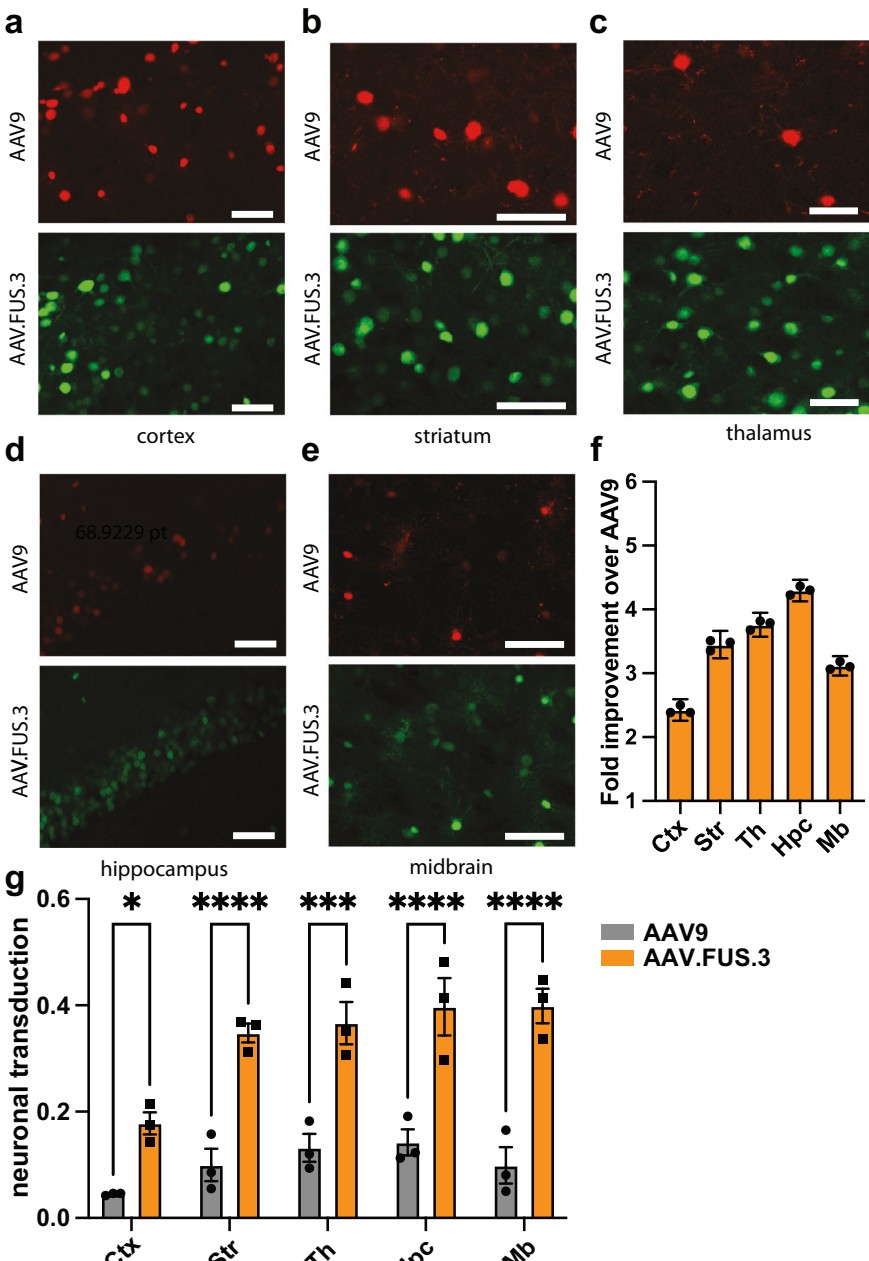

**Fig. 5 | AAV.FUS.3 shows regional dependence of transduction efficiency.** Hippocampus showed the highest, 4.3-fold, improvement in transduction over AAV9. **a** Representative image comparing transduction of the cortex with AAV.FUS.3 (green) and AAV9 (red). **b** Representative image comparing transduction of the striatum with AAV.FUS.3 (green) and AAV9 (red). **c** Representative image comparing transduction of the thalamus with AAV.FUS.3 (green) and AAV9 (red). **d** Representative image comparing transduction of the hippocampus with AAV.FUS.3 (green) and AAV9 (red). **e** Representative image comparing transduction of the midbrain with AAV.FUS.3 (green) and AAV9 (red). **f** AAV.FUS.3 shows regional differences in transduction efficiency of the tested regions – cortex (Ctx), striatum (Str), thalamus (Th), hippocampus (Hpc), midbrain (Mb). All differences were statistically significant (All pairwise comparison $p$-values < 0.0001, except thalamus vs striatum ($p = 0.0026$) and striatum vs midbrain ($p = 0.0015$), $n = 3$ mice per region, one way ANOVA, $F(4, 10) = 283.4$, $P < 0.0001$; Tukey HSD post-hoc test; center for the error bars represents arithmetic mean.). **g** Neuronal transduction efficiency for AAV9 (gray) and AAV.FUS.3 (orange). AAV.FUS.3 showed significant improvement over AAV9 transduction in all tested regions, $n = 3$ mice per region (two-way ANOVA with Sidak's test; $F(1, 20) = 141.2$; $p = 0.0333$, $p < 0.0001$, $p = 0.0002$, $p < 0.0001$, $p < 0.001$ for Cortex, Striatum, Thalamus, Hippocampus and Midbrain, respectively; center for the error bars represents arithmetic mean.). IV injection dose, $10^{10}$ VG/g. Scale bars are 50 μm. Error bars are 95% CI.

microglia/macrophages significantly more efficiently than AAV.FUS (3.5% and 0.7%, respectively; $n = 6$ sections analyzed from $n = 3$ mice, $p = 0.0174$, paired $t$ test, $t = 7.487$) as well as oligodendrocytes (74.3% and 3.4% respectively; $n = 18$ sections analyzed from $n = 6$ mice, $p < 0.0001$, paired $t$ test; $t = 12.32$). (Supplementary Fig. S7). These results show that in addition to improved specificity for targeted regions of the brain, the engineered viral capsids are more selective for neurons over other cephalic cell types.

Based on its leading combination of neuronal tropism and improvement in brain specificity among the engineered variants, we selected AAV.FUS.3 for further evaluation as a FUS-BBBO-specific viral vector.

## AAV.FUS.3 transduction at a low dose

Lowering the dose of AAVs during gene therapy or scientific studies is of high interest due to lower cost and reduced toxicity[43]. We decided to evaluate whether the improvements in transduction can be retained at lower dose, such as $10^9$ vg/g, which has been used in other FUS-BBBO gene delivery studies[44]. Our results showed that AAV.FUS can transduce the brain more efficiently than AAV9 at this dose, with a total number of transduced cells being $2.2 \pm 0.6$-fold higher for AAV.FUS over AAV9 ($n = 6$ mice analyzed, $p = 0.0004$, two-tailed paired $t$ test, $t = 8.182$; Supplementary Fig. S8a, b). At the same time, the liver transduction was lower for AAV.FUS compared to AAV9 by $5.2 \pm 1.6$-fold ($n = 6$ mice analyzed, $p = 0.0004$, two-tailed paired $t$ test; $t = 8.530$; Supplementary Fig. S8c, d), reaching the overall brain-to-liver transduction ratio of $11.6 \pm 3.7$ (95% CI, Supplementary Fig. 8e), which was comparable to the brain-to-liver transduction ratio at a higher dose of $10^{10}$ vg/g (12.1-fold vs 11.6-fold; $p = 0.798$; two-tailed, heteroscedastic $t$ test, $t = 0.2682$). Finally, we evaluated the overall neuronal transduction efficiency at this dose and found the average transduction measured across three brain regions (striatum, thalamus, hippocampus) of $12.6\% \pm 3.7\%$ for AAV9 and $54.4\% \pm 8.8\%$ for AAV.FUS, for a total of 4.6-fold difference ($p < 0.0001$; two-tailed paired $t$ test; $t = 14.81$; Supplementary Fig. S8f, g). Overall, the properties of AAV.FUS.3 for enhanced brain-specific transduction and neuronal tropism over AAV9 were retained at the lower vector dose.

## Region-specific transduction efficiency of AAV.FUS.3

To further characterize AAV.FUS.3's performance relative to AAV9, we decided to evaluate the efficiency of delivery when these vectors are targeted to different brain regions. To ensure that each region is targeted exclusively, only one brain region was targeted with FUS-BBBO in each tested mouse. To ensure the rigor of this investigation and account for variability in virus titration[45], we obtained a new batch of both AAV9 and AAV.FUS.3 and titered them independently. We evaluated the efficiency of transduction when these vectors were targeted by FUS-BBBO to the striatum (caudate putamen), thalamus, hippocampus, and midbrain.

We observed a major improvement in AAV.FUS.3 transduction compared to AAV9 in all targeted regions, with a fold-change ranging from $2.4 \pm 0.08$ to $4.3 \pm 0.08$ (95% CI, Fig. 5). Among brain regions, we found that the hippocampus (Hpc) is transduced with a particularly elevated relative efficiency while the cortex (Ctx) showed the lowest with a 2.4-fold improvement. These results indicate that AAV.FUS.3 can target multiple brain regions with improved efficiency, while suggesting the potential for further engineering AAVs with region-enhanced tropism in FUS-BBBO delivery. Lower magnification images showing transduction in surrounding brain areas can be found in Supplementary Fig. S10.

## AAV.FUS.3 transduction is improved over AAV9 after direct intraparenchymal delivery

Given its improved efficiency of neuronal transduction after FUS-BBBO, we hypothesized that AAV.FUS.3 may also show improved efficiency upon intraparenchymal injection. Such improvement in transduction would suggest that at least part of that effect is due to improved transduction efficiency once the AAV.FUS.3 enters the brain, rather than from improve rate of passage across the FUS-opened BBB. Indeed, when injected into the hippocampus, our evolved variant showed 2.29-fold increased transduction efficiency compared to AAV9, which is similar to the 2.56-fold improvement seen with FUS-BBBO (Supplementary Fig. S11).

## AAV.FUS.3 transduces the brain in a mouse strain in which it was not selected

Viral vectors engineered through high-throughput screening and selection can exhibit properties that are limited to the strain of animal in which they were selected[46]. To evaluate the versatility of AAV.FUS.3, we tested its brain and liver transduction in BALB/cJ mice. Our analysis demonstrated that the enhanced properties of AAV.FUS.3 that we observed in C57BL/6J were also present in the BALB/cJ mice. We found significantly improved brain transduction efficiency ($3.9 \pm 0.1$-fold on average across brain regions) (Supplementary Fig. S12a) while showing $4.1 \pm 0.3$-fold reduction in liver transduction compared to AAV9 in BALB/cJ ($n = 6$ mice tested, Supplementary Fig. S12b), for a total brain-to-liver transduction ratio of $16.1 \pm 0.9$-fold ($n = 6$ mice tested) which was higher than what we found in C57BL/6J mice (16.1-fold and 12.1-fold, respectively, $n = 6$ for each group; $p = 0.000376$, two-tailed heteroscedastic $t$ test, $t = 5.350$). Further analysis showed that AAV.FUS.3 has significantly higher neuronal tropism in BALB/cJ mice as well, with 73% ($\pm 2.2\%$, $n = 6$) total brain cell transduction identified as neurons ($p < 0.0001$, two-tailed paired $t$ test, $t = 21.48$, Supplementary Fig. S12d). Overall, in BALB/cJ mice we saw an increase in neuronal transduction across all the tested regions, similar to what was observed in C57BL/6J mice ($n = 4$–6 mice per region; p < 0.0001 for all tested comparisons, two-way ANOVA with Sidak's multiple comparison test, $F(1, 44) = 494.1$, Supplementary Fig. S12e). Similarly to C57BL/6J, in BALB/cJ mice, we observed the highest improvement in the transduction efficiency of AAV.FUS.3 over AAV9 in the hippocampus (Hpc) (4.3-fold, Supplementary Fig. S12e). Lower magnification images showing transduction in surrounding brain areas can be found in supplementary Fig. S13.

## Discussion

Our results show that viral vectors can be engineered to improve noninvasive, site-specific gene delivery to the brain using ultrasound-mediated blood-brain barrier opening. Gene therapy is widely used in research and is becoming a clinical reality. However, most of the available methods for gene delivery to the brain either lack regional specificity or are invasive and challenging to apply to large brain regions[4–7,19,20]. On the other hand, FUS-BBBO has been safely used for gene delivery in a number of studies with naturally-occurring AAVs[8,9,11,21,25,26], including large brain volumes throughout the brain[17,18]. However, the optimization of ultrasound parameters[8,47,48] and equipment[49–51] alone is unlikely to affect the peripheral transduction. Thus, improving efficiency and tissue specificity of gene delivery with newly engineered vectors could lower the cost of the virus production and reduce immune responses to the vectors[52], but also reduce non-specific transduction[39,53,54] of peripheral tissues and associated toxicity[38,55,56]. These improvements can facilitate the widespread use of FUS-BBBO and provide a strategy to generate other improved AAVs for FUS-BBBO delivery.

In this study, we approached the problem of improving FUS-BBBO gene delivery by engineering the viral vectors themselves. The resulting improvements include an increase in brain transduction per viral vector injected, a reduction in peripheral expression, and an increase in neuronal tropism. Among the selected 5 AAV.FUS candidates, four transduced target brain sites more efficiently than AAV9 while also lowering transgene expression in the liver in the same mice. Our top candidate, which we call AAV.FUS.3, demonstrated improved transduction in five different brain regions and an overall efficiency of targeting the brain, defined as the ratio of brain to liver (peripheral) transduction, improved 12.1-fold compared to AAV9. This improvement in tissue specificity is particularly important because peripheral transduction can lead to toxicity. For example, AAV-based gene therapy has been shown to induce dose-dependent liver toxicity in clinical trials[37,43]. Our results show that AAV.FUS3 maintains its improved targeted brain transduction and reduced liver transduction relative to AAV9 at a lower dose of virus. The absolute transduction level observed at $10^9$ vg/g suggests that this relatively low dose may be sufficient for brain transduction. Furthermore, the relative similarity in transduction levels between

$10^9$ and $10^{10}$ vg/g suggests that higher systemic doses of AAVs may result in diminishing returns, consistent with previous work[57]. Larger-scale studies will be needed to evaluate peripheral transduction and toxicity in all peripheral organs and peripheral nervous system such as dorsal root ganglia (DRG)[38,58] in large animal species before potential translation of AAV.FUS.3.

Our results suggest the need to investigate the mechanisms by which AAVs enter the brain after FUS-BBBO and what accounts for the differences in efficiency among serotypes. The prevailing understanding of FUS-BBBO mechanisms suggests that FUS loosens tight junctions in the vasculature, allowing molecules and nanoparticles such as AAVs to pass from the blood into the brain[59]. Within this framework, reductions in peripheral uptake (leaving more AAV to circulate) and reduced binding to extracellular matrix[60] could help certain serotypes enter through physically generated openings and reach neurons more efficiently. We also found that direct intraparenchymal co-injection of AAV.FUS.3 and AAV9 showed improved transduction of the former, suggesting that AAV.FUS.3 transduces brain cells more efficiently after reaching the brain parenchyma. A final potential contributing factor is any molecular change that FUS-BBBO could cause to the vascular endothelium, leading to more complex interaction changes between viral vectors and their target. Understanding these factors would enable additional future engineering and optimization of FUS-BBBO-based gene delivery.

With further studies, AAVs engineered for FUS-BBBO-based gene delivery may provide clinical benefits over existing serotypes. Naturally occurring AAV serotypes, such as AAV9, have been successfully used in clinically approved therapies[61–64], including AAV9 intravenous delivery at doses higher than presented in this study[61]. Recently, a groundbreaking study has also shown that delivery of AAV9 into the brain can be achieved in non-human primates using FUS-BBBO[65], further bolstering the translational potential of this procedure. The current limitations of gene therapies include commonality of pre-existing neutralizing antibodies in a large fraction of the population[66]; high liver transduction leading to toxicity[43] and potential carcinogenicity[67]; AAV-induced toxicity in DRGs in primates[38,58]; and high cost of the therapy[68]. At least some of these problems could be addressed with viral vector engineering for improved brain transduction after FUS-BBBO. We expect viral capsids engineered under our paradigm can be instrumental in facilitating both pre-clinical and clinical studies. To make such engineered AAVs translatable, the major future challenge remains to identify which of these engineered vectors will be efficacious in humans.

Overall, this study shows that the molecular engineering of AAV capsids can lead to improved ultrasound-mediated gene delivery to the brain. Our screen yielded AAV.FUS.3, the first, to the best of our knowledge, viral vector expressly engineered to work in conjunction with a specific physical delivery method.

## Methods

### Animals
Animals. 10–14 week-old C57BL/6J, BALB/cJ, and Syn-1-Cre mice were obtained from Jackson Lab. Both male and female mice were used in the study, as described in the source data file. Animals were housed in a 12 h light/dark cycle and were provided with water and food ad libitum. All experiments were conducted under a protocol approved by the Institutional Animal Care and Use Committee (IACUC) of the California Institute of Technology and Rice University.

### Focused ultrasound equipment and BBB opening procedures
FUS-BBBO. Syn1-Cre, C57BL/6J, and BALB/cJ mice (10–14 weeks old) were anesthetized with 2% isoflurane in air, the hair on their head removed with Nair depilation cream and then cannulated in the tail vein using a 30-gauge needle connected to PE10 tubing. The cannula

was then flushed with 10 units (U)/ml of heparin in sterile saline (0.9% NaCl) and attached to the mouse tail using tissue glue (Gluture). Subsequently, the mice were placed in the custom-made plastic head mount and imaged in a 7 T MRI (Bruker Biospec). A fast low-angle shot sequence (echo time TE = 3.9 ms, repetition time TR = 15 ms, flip angle 20°) was used to record the position of the ultrasound transducer in relation to the mouse brain. Subsequently, the mice were injected via tail vein with AAVs. Within two minutes after viral injection, the mice were also injected with $1.5 \times 10^6$ DEFINITY microbubbles (Lantheus) and 0.125 μmol of ProHance (Bracco Imaging) dissolved in sterile saline, per g of body weight. The dose of DEFINITY was identical as used in our previous studies[1]. The dose of ProHance was chosen based on the manufacturer's recommendations. Within 30 s, the mice were insonated using an eight-channel FUS system (Image Guided Therapy) driving an eight-element annular array transducer with a diameter of 25 mm and a natural focal point of 20 mm, coupled to the head via Aquasonic ultrasound gel. The gel was placed on the top and both sides of the animal's head to minimize reverberations from tissue/air interfaces. The focal distance was adjusted electronically. The ultrasound parameters used were 1.5 MHz, 1% duty cycle, and 1 Hz pulse repetition frequency for 120 pulses and were derived from a published protocol. The pressure was calibrated using a fiber optic hydrophone (Precision Acoustics), with 21 measurements and uncertainty of ±3.8% (SEM). The pressure for FUS-BBBO was chosen to maximize the safety of delivery and was chosen on the basis of our previous studies[1] and preliminary data in our laboratory. The ultrasound parameters were 1.5 MHz, 0.33 MPa pressure accounting for skull attenuation (18%)[69], 1% duty cycle, and 1 Hz pulse repetition frequency for 120 pulses. For each FUS site, DEFINITY and Prohance were re-injected before the additional insonation. Each animal underwent four insonations located in one hemisphere, starting from the midbrain and going forward. The time between each insonation was approximately 3 min and included 120 s of insonation and 1 min for readjustment of positioning on the stereotaxic frame. The center focus of beams was separated by 1.35–1.5 mm (depending on mouse weight 25–35 g) in the anterior / posterior direction.

For the low-dose AAV.FUS.3 evaluation, we used different equipment since the original setup became unavailable. For this study, we used the RK50 (FUS Instruments) with the same center frequency (1.5 MHz) and $f = 0.7$. We used the same pulse length, frequency, and number of pulses as before. Since the pressure calibration shows high variation (±20% for fiber optic hydrophone used in this study; Precision Acoustics, Dorchester, UK) we adjusted the voltage on the transducer empirically to match the previous experiment and provide BBB opening without tissue damage (input peak-to-peak voltage of 14.2 V, corresponding to the peak negative pressure of 0.52 MPa when calibrated against the original transducer using a needle hydrophone (Onda)). Instead of MRI guidance, we used bregma-lambda targeting. Briefly, the mouse was mounted on a stereotactic platform using ear bars, bite bar and nose cone. A midline scalp incision was vertically made to expose the skull after disinfecting the site using three alternating scrubs of chlorhexidine scrub and chlorhexidine solution. Bregma-lambda locations were then registered in the RK50 software using a guide pointer, and FUS-BBBO was carried out as described above.

### Plasmids and DNA library generation
The plasmids used were either obtained from Addgene, Caltech's vector core, or modified from these plasmids. The AAV library genome used for selection (acceptor plasmid, rAAV9Rx/a-delta-CAP) was obtained from Caltech's vector core facility, as were other plasmids (REP2-CAP9Stop-DeltaX/A, pUC18). The Rep-Cap plasmid for packaging AAV.FUS candidates were modified from Addgene plasmid #103005 by introducing mutations selected from the screen. For testing the transduction, we used a plasmid obtained from Addgene

(pAAV-CaG-NLS-EGFP - #104061) and a plasmid modified in-house with exchanged EGFP for mCherry protein (pAAV-CaG-NLS-mCherry).

Mutations were introduced into the acceptor plasmid using a PCR with degenerated primers (7MNN) with a sequence 5′-GTATTCCT TGGTTTTGAACCCAACCGGTCTGCGCCTGTGCNMNNMN NMNNMNN MNNMNNTTGGGCACTCTGGTGGTTTGTG-3′, targeted as a 7-aminoacid insertion between residues 587 and 588. The amplified insert was then introduced into the capsid plasmid through restriction cloning using XbaI and AgeI enzymes. DNA from the treated brain was recovered by PCR using two pairs of plasmids – the first step of amplification was done using 5′-CAGGTCTTCACGGACTCA-GACTATCAG-3′ and 5′-CAAGTAAAACCTCTACAAATGTGGTAAAATCG-3′ primers which selected for the DNA that has been modified by Cre enzyme. The second stage, intended to amplify the DNA was performed using a pair of primers: 5′-ACTCATCGACCAATACTTGTACT ATCTCTCTAGAAC-3′ and 5′- GGAAGTATTCCTTGGTTTTGAACCC AA-3′.

## Virus production and purification
AAV library was purified as previously published[6]. In short, we transfected the DNA carrying a genome containing capsid which has been modified by the 7-mer insertion (10 ng per 100 mm diameter dish), the helper DNA containing REP protein (10 μg per 100 mm diameter dish, and 9.99 μg of empty pUC19 carrier plasmid), and an AdV helper plasmid (20 μg per 100 mm diameter dish) using PEI. Media was changed 16 h after transfection, and then collected 48 h post-transfection and stored in 4 C. 60 h after the transfection, we scraped the cells into San digestion buffer (Tris pH 8.5 with 500 mM NaCl and 40 mM MgCl2 with Salt Active Nuclease). Virus in the media was precipitated using 1/5 volume of 5X PEG8000+NaCl (40% PEG-8000 and 2.5 M NaCl), incubated on ice for 2 h, and spun at $3000g$ for 30 min at 4 C. The media and cell-scraped stocks were then combined and precipitated using iodixanol gradient precipitation (virus appears on the 40−60% iodixanol interface), diluted into 15 ml PBS with 0.001% Pluronic-F68, and sterile-filtered through a 0.2-μm PES filter. Finally, the buffer was dialyzed using Amicon 100 KDa cut-off centrifuge filters at least 3 times to remove residual iodixanol, after which the virus was tittered using a standard qPCR protocol[6] (Vigene Biosciences, Rockville Maryland). All batches of the AAV were purchased from the same company and the same production batch was used for co-administration of AAVs. AAV.FUS candidates were packaged and titered using a commercial service (Vigene biosciences) to ensure reproducibility for external investigators, as the titers can show variability between different labs[45]. We have re-titered the AAV.FUS.3 and AAV9 from another batch again in our lab, to make sure that the improvement of AAV.FUS over AAV9 is consistent between investigators.

## In vivo selection and gene delivery
To enable in vivo selection of AAV.FUS we delivered the AAV library to one hemisphere through FUS-BBBO. We targeted four sites corresponding to the striatum, dorsal hippocampus, ventral hippocampus, and midbrain using MRI guidance. We used 0.33 MPa pressure and other parameters as described in the *Focused ultrasound equipment and BBB opening procedures* section. The parameters used were identical during the in vivo selection and testing of the AAV.FUS candidates. The AAVs were delivered intravenously. For the first round of selection, the dose delivered was 6.7E9 viral genomes per gram of body weight. The library for the first round of evolution contained 1.3E9 sequences, yielding approximately 5 genomes of each clone per gram of body weight. For the second round, where the library contained 2098 candidates, 1.3E9 viral genomes per gram of body weight were delivered, yielding 6.2E5 viral genomes for each clone per gram of body weight. Following the selection of a single candidate (AAV.FUS.3) for further analysis,

we repeated the above procedure using a dose of 1E9 viral genomes per gram of body weight. Following FUS-BBBO or intraparenchymal injections, mice were returned to the home cages for 14 days, after which they were euthanized by $CO_2$ overdose.

## Intraparenchymal injections
Using a stereotaxic frame (Kopf), intraparenchymal co-injections of AAV9 and AAV.FUS.3 were also performed using a microliter syringe equipped with a 34-gauge beveled needle (Hamilton) that is installed to a motorized pump (World Precision Instruments). Each AAV was injected unilaterally at a dose of 4E8 viral genomes per gram of body weight to the CA1 in the hippocampus (AP −1.94 mm, ML + 1.0 mm, DV −1.3 mm) infused at a rate of 200 nL/min, and the needle was kept in place for 5 min before removing it from the injection site.

## Tissue preparation for DNA extraction
The brains of mice euthanized with $CO_2$ overdose were extracted, and the targeted hemisphere was separated from the control hemisphere with a clean blade. Each hemisphere was then frozen at −20C prior to DNA extraction. The brains were then homogenized in Trizol using a BeadBug tissue homogenization device with dedicated pre-filled 2.0 ml tubes with beads (Zirconium coated, 1.5 mm, Benchmark Scientific, Sayreville, New Jersey) for 1−3 min until tissue solution was homogenous. The DNA was then extracted with Trizol and amplified first with CRE-independent, and then CRE-dependent PCR, first through 15−25 cycles, and then 15 cycles of PCR[6] with Q5 Hot-Start DNA polymerase using the manufacturer's protocols (NEB, Ipswich, MA). For the first step of PCR amplification we used 5′-CAGGTCTTCACG-GACTCAGACTATCAG-3′ as a forward primer, and 5′-CAAG-TAAAACCTCTACAAATGTGGTAAAATCG-3′ as a reverse primer. For the second step, we used 5′-ACTCATCGACCAA-TACTTGTACTATCTCTCTAGAAC-3′ as a forward primer, and 5′-GGAAGTATTCCTTGGTTTTGAACCCAA-3′ as a reverse primer.

## Next generation sequencing data analysis
The variable region of all detected capsid sequences was extracted from raw fastq files using the awk tool in Unix terminal. This process filtered out sequences not containing the constant 19 bp region flanking each side of the variable region. Sequences were then sorted, checked for length, and ordered from highest to lowest copy number in the sequencing experiment. During the first screen, the top 3000 were chosen. Among these 3000, any sequence that was only a point mutation away from a sequence and 30x less abundant was removed and assumed to be a potential sequencing readout error. This led to our final library of 2098 sequences, which were synthesized by Twist Biosciences (San Francisco, CA) for use in the second round of screening. This second AAV library also included a set of 2098 "codon-optimized" capsid variants that were encoded for the same protein as the original sequences but using a different DNA sequence chosen by the IDT codon optimization tool. To process the second batch of sequencing data, we first normalized the copy numbers of the sequences in each experiment to one to ensure the comparability of different samples. Then, we filtered out sequences that were not contained within the input library. Finally, we evaluated the normalized frequency of reads for each sequence, defined as the normalized copy number of each sequence averaged among original and codon-optimized variants for each capsid. Top sequences for further analysis were selected to be the most abundant sequences that appeared at least 100x more frequently in the targeted brain hemisphere than the non-targeted hemisphere in all tested mice, and from these sequences, the top 5 were chosen as AAV.FUS candidates.

## Histology, and image processing
After cardiac perfusion and extraction brains were post-fixed for 24 h in neutral buffered formalin (NBF). Brains were then sectioned

coronally at 50-μm on Compresstome VF-300 (Precisionary Instruments, Natick, MA). Sections were immunostained with anti-NeuN Alexa Fluor 405-conjugated antibody (1:500 dilution, RBFOX3/NeuN Antibody by Novus Biological, stock number: NBP1-92693AF405), anti-GFAP Alexa Fluor 405-conjugated antibody (1:500 dilution, GFAP Antibody by Novus Biological, stock number: NBP1-05197AF405), and anti-Iba1 Alexa Fluor 405-conjugated antibody (1:500 dilution, Iba1 Antibody by Novus Biological, stock number: NBP1-75760AF405). For oligodendrocyte staining, sections were immunostained with rabbit anti-Olig2 antibody (1:200 dilution, Abcam, stock number: 109186) and Alexa Fluor 647-conjugated goat anti-rabbit IgG antibody (1:200 dilution, Invitrogen, stock number: A21244). Sections were imaged on a Zeiss LSM-800 microscope using a 20x objective. Channels' laser intensities normalized to the brightness of mCherry and GFP proteins, the fluorescence of which was used to evaluate transduction. Images were then randomized, and anonymized. The experimenter was blinded in terms of fluorophore color, tested AAV strain, or the mouse identification (H.L., M.H). One data set (Fig. S7) was not anonymized due to the error in file-sharing setting. Three 50-μm coronal sections of the brain were analyzed for each mouse, for each strain of the AAV including the section at the center of the FUS-target and the sections 500 and 1000 μm anterior to that section.

The FUS-BBBO-targeted regions for evaluation of transduction efficiency and the total transduced cells were selected by setting the regions of interest (ROI) to be the area bound by gene expression at the FUS-targeted site due to AAV transduction such that less than 1% of all transduced cells are outside the bounds of the ROI.

The data was then independently validated by an experimenter blinded to the goals of the study (J.T). The inter-experimenter variability was 12.5% (1.9-fold (RL, primary scorer) vs 2.1-fold difference (JT, secondary scorer), $n = 15$ randomly selected images, a total of 11,230 cells counted) and the difference between the scores was not statistically significant ($p = 0.071$, two-tailed, paired $t$ test). To evaluate the BBB permeability of the AAV in the absence of FUS-BBBO (off-target transduction), a randomly chosen untargeted region at least 2 mm from the center of the targeted region (4 times the distance of distance half-width half maximum of pressure, resulting in ~16-fold pressure reduction) was used within the same sections that were used to evaluate transduction efficiency at FUS focus.

### Statistical analysis

A two tailed $t$ test, without assuming equal variance, was used when comparing the means of two data sets. For the comparison of more than two data sets, one-way ANOVA was used, with Tukey's HSD post-hoc test to determine the significance of pairwise comparisons. When more than one variable was compared across multiple samples, two-way ANOVA was used, followed by Sidak's multiple comparisons test with F statistic provided with regards to differences between AAV9 and AAV.FUS candidates. Pairing was used if the compared data was obtained from the same specimen (e.g. histological analysis after co-injection of two viral vectors into a single mouse), and not used if the data were analyzed from different unrelated specimens. Specific $p$-values are provided in a source data appendix, due to the large numbers of pairwise comparisons in this study. Software (Prism 9) was used for statistical analysis, and the minimum $p$-value calculated by the software was $p < 0.0001$.

### Reporting summary

Further information on research design is available in the Nature Portfolio Reporting Summary linked to this article.

## Data availability

The authors declare that all data supporting the results in this study are available within the paper, its Supplementary Information and its Source Data file. Microscopy images and raw sequencing data are available from the corresponding author upon reasonable request owing to their large size and numbers. The NGS data generated through this study have been deposite in the Sequence Read Archive (SRA) database under accession code: PRJNA1112439. Source data are provided with this paper.

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

## Acknowledgements

The authors thank Drs. Benjamin Deverman, Nicholas Flytzanis, Nicholas Goeden, and Viviana Gradinaru, and the CLOVER center at Caltech for helpful discussions and the Biological Imaging Facility of the Beckman Institute. This research was supported by the National Institutes of Health (grant UG3MH120102 to M.G.S.), the Jacobs Institute for Molecular Engineering in Medicine, the Sontag Foundation, the Merkin Institute for Translational Research, and 2019 NARSAD Young Investigator Grant from the Brain and Behavior Research Foundation (grant 27737 to J.O.S.). Related work in the Shapiro Lab is supported by the David and Lucille Packard Foundation and the Heritage Medical Research Institute and in Szablowski lab by The G. Harold and Leila Y. Mathers Charitable Foundation. M.G.S. is an Investigator of the Howard Hughes Medical Institute. JEH acknowledges support from Rose Hills foundation and Barry Goldwater Scholarship and from the NSF GRFP. MH acknowledges support from NSF GRFP.

## Author contributions

H.L., J.O.S., and M.G.S. conceived and planned the research. H.L., M.H., and J.O.S. performed the in vivo experiments with additional input from J.E.H. J.O.S., H.L., and J.E.H. performed the in vitro experiments. J.E.H. and J.O.S. processed the next-generation sequencing data. H.L., M.H., and J.S.T. processed and analyzed histological image data. J.O.S. and M.G.S. wrote the manuscript with input from all other authors. M.G.S. and J.O.S. supervised the research.

## Competing interests

J.O.S., M.G.S., J.E.H., and H.L. are inventors on the patent application US20230047753A1. Other authors declare no competing interests.
