## [Peer Review File · Nature Communications]

Reviewers' Comments:

Reviewer #1:

Remarks to the Author:

Title: Engineering viral vectors for acoustically targeted gene delivery

Until now, AAV-based gene therapy for brain targeting remains the big challenge because of the existence of physical barrier, blood-brain barrier. Numerous efforts have been focused on the development of engineered AAV vectors. The manuscript by Hongyi Li et al. screened and identified multiple AAV9 variants by in vivo selection by intravenous injection in conjunction with non-invasive FUS-BBBO method. The resulting variants could substantially enhance the neuronal tropism while reducing peripheral transduction (data in liver). The authors explored a new non-invasive method to surmount the challenge of brain delivery. However, there are several major concerns about the study as below.

1 For brain-targeted AAV-based gene therapy, it is important and necessary to measure the AAV transduced efficacy in various brain cells (such as neuron cells, endothelial cells, and astrocytes cells etc.). The whole study here just described the AAV variants transduced efficiency compared to AAV9 in local neuron cells. Did the authors also measure the transduced efficiency in other brain cells?

2 Following the 1st question, did the author observe AAV variants transduction change in other peripheral organs except the liver?

3 In supplementary Fig S3 and Fig S6, the authors did the pairwise comparison. What is the main goal of the results? Both are repeated with Fig 3b and Fig 5f, respectively. I do not think these data are necessary for the whole story.

4 In Fig 1a, it will be better to merge the neuron cells and AAV transgene expression (mCherry and GFP expression by AAV9 and AAV.FUS.3, respectively).

5 In Fig. 4 b and c, the quality of representative images is poor and hard to exactly differentiate neuron cells and non-neuron cells. More amplified imaging should be employed to optimize the images.

6 The manuscript should be polished to better reading.

Reviewer #2:

Remarks to the Author:

Li et al describe the development of novel AAV vectors optimized for delivery to the brain using focused ultrasound (FUS) and microbubbles to increase blood-brain barrier permeability. The engineered AAVs are screened and selected using an evolution-based method in mice. Five AAVs are selected based on their increased transduction of neurons. AAV.FUS.1-5 are hereafter compared to AAV9. Compared to AAV.FUS.3 shows increased brain transduction, decreased liver transduction, and increased neuronal tropism.

The idea of developing AAVs that are optimized for different delivery methods is relevant and of high interest to the field. With the novel vectors, the authors seek to overcome limitations for FUS-mediated delivery of AAVs including the high intravenous doses needed, the off-target transduction of peripheral organs, and the ultrasound parameters applied for AAV delivery.

The manuscript is interesting. However, it is short of evidence for several claims and many limitations must be addressed with additional experiments and modifications of the text.

Major concerns

1) A significant limitation in the study is the evolution-based approach that was used for screening new AAV candidates. This methodology will select AAV candidates with improved properties for the species from which they were selected; as reported, for example, for the AAV-PHP.B vector. This limitation can be a major impediment to applications in other strains of mice and other species.

1a) Experiments in other strains and species would strengthen the usefulness of AAV.FUS.3. What

are the transduction efficiencies tested in other strains and species?

1b) The text should acknowledge the advantages and disadvantages of AAV9 based on its track record in clinical trials and put it in perspective with AAV.FUS.3

2) The authors state several times in the manuscript that safety is a concern with the FUS parameters used in other studies for the delivery of AAVs, and that previous optimizations of FUS parameters and equipment have not solved these issues. This is not accurate. The detailed FUS parameters and microbubbles (type and concentration) used for the delivery of AAVs differ between research groups; however, many are within safety margins.

2a) A detailed description (e.g. Table) of the FUS parameters and microbubbles (type and concentration) that have been used successfully – and without safety concerns – to delivered AAVs to the brain with FUS should be provided.

2b) Previous FUS approaches and parameters that are raising safety concerns should be highlighted and discussed by the authors.

2c) Data on other peripheral organs and DRG would be important to the safety and peripheral transduction claims of the AAV.FUS.3 vector.

3) In the introduction, the authors hypothesize that three limitations can be overcome by developing novel AAV serotypes optimized for FUS-mediated delivery. One limitation outlined is the intravenous dose needed for AAV delivery to the brain using natural serotypes.

3a) While a 2-4 fold increase in transduction efficiency is significant in the current experiments, the authors should demonstrate whether this increase is high enough to substantially decrease the intravenous dose of AAV.FUS.3 that is required to reach relevant levels of transduction and expression. Specifically, can AAV.FUS.3 be administered intravenously at less than 10^{10} and still achieve high transduction efficacy with FUS?

b) The authors incorrectly report that AAVs are used with FUS in the order of 10^{10} . Several studies have shown that 10^9 VG/g was sufficient for AAVs delivery with FUS. How do the results of the present study compare?

c) What is the ability of the AAV9 (and AAV.FUS.3) to cross the BBB at non-FUS sites? At 10^{10} VG/g several studies have shown that AAV9 crosses the BBB. Please put in perspective with other studies.

4) It is intriguing that AAV.FUS.3 may be a vector that is specifically optimized for FUS delivery. The authors suggest that it could be caused by either increased AAV.FUS.3 levels in circulation because of lower binding to other tissue; however, only the liver was investigated in peripheral organs.

4a) Please show evidence of transduction (or lack of transduction) in other peripheral organs, DRG, and/or the residual levels of AAV.FUS.3 in circulation.

5) The authors also suggest that AAV.FUS.3 may lead to sophisticated changes in receptor binding and BBB interaction facilitates the crossing of the BBB at FUS-targeted sites. This could represent a future study of interest. Another option, which the authors did not mention, is the possibility that AAV.FUS.3 is better than AAV9 in transducing neurons upon brain entry.

5a) This can easily be tested by comparing AAV9 and AAV.FUS.3 transduction following intracranial injections. Adding this data will significantly contribute to the understanding of whether AAV.FUS.3 is specifically improved for FUS-mediated delivery or improved for neuronal transduction overall.

6) The neuronal tropism of AAV.FUS.3 and its comparison with AAV9 represent several pitfalls that must be addressed.

6a) While AAV.FUS.3 may have enhanced neuronal tropism, AAV.FUS.3 transduced liver cells and expressed GFP. Additional data on the cellular tropism and expression of AAV.FUS.3 in peripheral organs, DRG, and in the brain (neuronal, glial and others) are important.

6b) Data should be presented and compared in absolute numbers for AAV9 and AAV.FUS.3 (and not percentage or fold-improvement). Since AAV9 and AAV.FUS.3 may transduce neurons and glia differently depending on the brain regions, their comparison for neuronal tropism (e.g. Fig 4) and across brain regions (e.g. Fig 5) as percentages and fold improvement is not always the most informative.

Examples (there are others) requiring improvements:

6b.1) The enhanced neuronal tropism of AAV.FUS.3 in 4c seems to be limited to a dense neuronal cell layer. Brain regions and subregions should be identified clearly and analyzed separately if heterogeneity is present. What are the other cell types transduced by AAV9 and AAV.FUS.3? Is this the hippocampal formation, and if yes were exactly?

6b.2) In Figure 5, the important data to most readers may not be the fold improvement over AAV9 but the representative transduction efficiency for each vector, within each region. The raw data would show this nicely instead of fold-improvement (fold-improvement is interesting and can be mentioned in the text instead). For example, looking at the fold-improvement data (5f), one may think that AAV.FUS.3 is not great for cortical transduction but, judging from 5a one can think that both AAV9 and AAV.FUS.3 seems to transduce cortical cells better than hippocampal cells. Co-localization with NeuN and other cell types would be very informative in Figure 5.

Minor concerns

7) The authors state on lines 50-51 that AAVs can be safely delivered to small and large areas using FUS. The authors should modify this statement, as this is only known for FUS treatment alone and not in combination with AAV. Though no gross tissue damage has been reported following FUS-mediated delivery of AAVs to the brain, most studies are not long-term, and especially the immune reactions that may arise following AAV delivery do not cause tissue damage within those timeframes.

8) It is unclear if Supp Fig S2 is in the brain or liver? Since tissue autofluorescence differs between organs and fluorophores, the reliability of the quantification method using GFP and mCherry should be included for both brain and liver.

9) Lines 241-244 the authors state that FUS loosens tight junctions and allows substances to pass from the blood to the brain. The authors should also mention other mechanisms, e.g., increased transcytosis and decreased levels of efflux transporters.

10) Please clarify the brain processing and analysis:

10a) Brains were sectioned at 50-micron but 100-micron sections were analyzed?

10b) Which direction was chosen for brain sectioning? Horizontal, coronal or sagittal?
Since the FUS spots only target some parts of the brain, the sectioning and sampling of sections for analysis will have a large impact on the data obtained.

10c) Which area was quantified? Were all positive cells in the entire section counted or only in the FUS targeted area?

11) Since evolution-based methods have previously shown the development of AAVs with species-specific properties, please include in the mouse description the strain of origin.

Dear reviewers,

Please find the attached responses to your comments and questions. The responses are inset to improve readability. The changes to the manuscript are indicated in yellow in the main manuscript file.

Reviewer 1

1. For brain-targeted AAV-based gene therapy, it is important and necessary to measure the AAV transduced efficacy in various brain cells (such as neuron cells, endothelial cells, and astrocytes cells etc.). The whole study here just described the AAV variants transduced efficiency compared to AAV9 in local neuron cells. Did the authors also measure the transduced efficiency in other brain cells?

Thank you for this suggestion. We addressed this comment with a number of additional experiments. In addition to neurons, we now counterstained for astrocyte/glia (GFAP) and microglia (Iba1). The transduction is reported **in lines 218-222** and in **Supplementary Fig. S7**).

2. Following the 1st question, did the author observe AAV variants transduction change in other peripheral organs except the liver?

Following this suggestion, in addition to the liver, in this revision we sectioned and imaged kidneys and lungs and found low levels of transduction (several cells in 320-320 micron field of view) for both viral vectors was observed (in-text lines **193-198**; **Supplementary Fig. S5**).

3. In supplementary Fig S3 and Fig S6, the authors did the pairwise comparison. What is the main goal of the results? Both are repeated with Fig 3b and Fig 5f, respectively. I do not think these data are necessary for the whole story.

We added it to the supplementary information for the convenience of the readers. Tracking pairwise comparisons between multiple groups is easiest to follow on a graph rather than the table of p-values, but the figure was too large to be readable within the context of other panels in the main text figure.

4. In Fig 1a, it will be better to merge the neuron cells and AAV transgene expression (mCherry and GFP expression by AAV9 and AAV.FUS.3, respectively).

We assume the reviewer is referring to Figure 3a since Fig. 1a is an outline of the screening methodology. We represented the neurons in separate panels due to a significant fraction of population having various types of color blindness and separating the panels is more readable to these groups of people.

5. In Fig. 4 b and c, the quality of representative images is poor and hard to exactly differentiate neuron cells and non-neuron cells. More amplified imaging should be employed to optimize the images.

Thank you for this comment. We revised the figure to use larger magnification of images with higher quality imaging in **Fig. 4b, c**. We also added new data in the supplement showing co-staining with non-neuronal cell-types and quantified the AAV.FUS and AAV9 specificity in **Supplementary Figure S7**.

6. Polished for better reading;

The manuscript has been substantially rewritten; all changes are highlighted in yellow.

Reviewer 2

1) A significant limitation in the study is the evolution-based approach that was used for screening new AAV candidates. This methodology will select AAV candidates with improved properties for the species from which they were selected; as reported, for example, for the AAV-PHP.B vector. This limitation can be a major impediment to applications in other strains of mice and other species.

1a) Experiments in other strains and species would strengthen the usefulness of AAV.FUS.3. What are the transduction efficiencies tested in other strains and species?

We appreciate this comment. Based on this suggestion, we repeated the experiment in BALB/cJ mice and found AAV.FUS.3's properties were retained in BALB/cJ. We found AAV.FUS.3, when compared to AAV9, improved transduction of efficiency in the brain, reduced transduction in the liver, and showed improved neuronal tropism. Overall brain-to-liver improvement of AAV.FUS.3 over AAV9 was even higher for BALB/cJ than for C57BL/6J, at 16.1-fold vs 12.1-fold improvement, respectively. The data is presented in **Supplementary Figure S11** and in text **lines 262-285**.

We hypothesize that a reason our selected vectors are more transferrable across strains compared to PHP.B is the different mechanism of transport across the BBB. In PHP.B, the BBB crossing is dependent on binding to a specific receptor on the vascular endothelium (specifically the Ly6 receptor, which is absent in some strains of mice, including BALB/cJ). In contrast, our capsid variants were selected to enter the brain after FUS-BBBO, in which the BBB is thought to be compromised through mechanical disruption, which is not expected to be dependent on a specific endothelial receptor. The improved transduction observed with AAV.FUS likely arises from more generally improved pharmacokinetic and transport properties and increased transduction of neurons after entering the brain. Relevant discussion on the AAV.FUS mechanism is provided in the discussion in **lines 320-323**.

The investigations of AAV.FUS performance in other species is currently a part of another manuscript being written for submission.

References:

Hordeaux J et al, Mol Ther. 2018 Mar 7; 26(3): 664–668.

1b) The text should acknowledge the advantages and disadvantages of AAV9 based on its track record in clinical trials and put it in perspective with AAV.FUS.3

Thank you. We added more discussion and citations to studies using AAV9, including in clinical trials, in lines 74-75 and 101-103. In addition, we provided a brief description of challenges facing naturally-occurring and engineered AAVs in potential clinical translation, **in lines 327-335.**

2) The authors state several times in the manuscript that safety is a concern with the FUS parameters used in other studies for the delivery of AAVs, and that previous optimizations of FUS parameters and equipment have not solved these issues. This is not accurate. The detailed FUS parameters and microbubbles (type and concentration) used for the delivery of AAVs differ between research groups; however, many are within safety margins.

We agree with this comment and have amended the discussion to state that FUS-BBBO can be done safely, and that our goal is to enable viral delivery with FUS-BBBO to operate even further within the safety margin of FUS parameters to facilitate widespread use, **in lines 294-301.**

2a) A detailed description (e.g. Table) of the FUS parameters and microbubbles (type and concentration) that have been used successfully – and without safety concerns - to delivered AAVs to the brain with FUS should be provided.

We provided a citation for a recent review paper that lists those parameters in **line 294** (ref. 26).

2b) Previous FUS approaches and parameters that are raising safety concerns should be highlighted and discussed by the authors.

Added on **lines 293-294.**

2c) Data on other peripheral organs and DRG would be important to the safety and peripheral transduction claims of the AAV.FUS.3 vector.

In addition to the liver, we sectioned and imaged kidneys and lungs, and found no substantial transduction at the doses used (several cells per field of view at 320-320 microns; in-text **lines 193-198**; Supplementary **Fig. S5**).

Immunostaining of DRGs was not possible in our case without substantial background, which prevented identification of cells when using tissue clearing methods, and sectioning in vibratome was unsuccessful due to the small size of mouse DRGs (~1mm-sized tissue pieces after extraction). We have indicated the need for this additional testing in future studies on lines **312-313**, and **327-335**.

3) In the introduction, the authors hypothesize that three limitations can be overcome by developing novel AAV serotypes optimized for FUS-mediated delivery. One limitation outlined is the intravenous dose needed for AAV delivery to the brain using natural serotypes.

3a) While a 2-4 fold increase in transduction efficiency is significant in the current experiments, the authors should demonstrate whether this increase is high enough to substantially decrease the intravenous dose of AAV.FUS.3 that is required to reach relevant levels of transduction and expression. Specifically, can AAV.FUS.3 be administered intravenously at less than 1×10^{10} and still achieve high transduction efficacy with FUS?

We agree that improving transduction efficiency at a lower dose of AAV is important. To address this, we evaluated the transduction efficiency of AAV.FUS.3 and AAV9 at 1×10^9 vp/g ($1/10^{\text{th}}$ of the original dose) and found AAV.FUS.3 showed a greater, 4.3-fold improvement in the neuronal transduction efficiency vs AAV9, at 54.4% vs 12.6%, respectively. These new results are reported **in lines 228-245 and Supplementary Figure S8**.

b) The authors incorrectly report that AAVs are used with FUS in the order of 1×10^{10} . Several studies have shown that 1×10^9 VG/g was sufficient for AAVs delivery with FUS. How do the results of the present study compare?

We clarified the sentence in question to indicate the ranges used in the literature and cited the relevant papers in lines 65 to 70. We provided citations showing the range of doses used in the published studies with FUS-BBBO AAV delivery ranging from 5×10^8 to 1.67×10^{10} viral particles per gram of body weight, reported in **lines 72 and 76**.

Regarding the comparison, a recent comprehensive study of transduction of naturally occurring serotypes was published after this manuscript was submitted, which shows that 3.3×10^9 vp/g of AAV9 lead to transduction of up to ~6% neurons, higher than any other tested naturally-occurring serotype (AAV1, AAV2, AAV5, AAV8; Kofoed et al (2022)).

In terms of comparison, we designed our study to enable a head-to-head comparison of AAV.FUS with AAV.9 within the same individual animals, which we believe provides the best-controlled measure of relative efficiency.

c) What is the ability of the AAV9 (and AAV.FUS.3) to cross the BBB at non-FUS sites? At 10^{10} VG/g several studies have shown that AAV9 crosses the BBB. Please put in perspective with other studies.

Thank you for this question. None of the AAV.FUS candidates produced substantial off-target expression within brain at sites not insonated by FUS, as quantified in **Fig. 3c** and shown visually in **Fig. 3a**. We found 0.35% neuronal transduction for AAV9 and 0.17% for AAV.FUS.3. Thus, while it is possible to achieve transduction without BBB opening, the efficiency is approximately ~ 2 orders of magnitude lower (See **Fig. 5g**) than with the BBB opening.

Studies to date have similarly shown propensity of AAV9 at doses 5×10^8 to 1.25×10^{10} vp/g to transduce in the insonated areas and not the contralateral sites, including, for example the studies referenced in the manuscript: Thevenot et al (2012), Hsu et al. (2013), Wang et al (2015), Szablowski et al (2018), Kofoed et al (2022), Nouraein et al (2023). Specifically, the latter study was done by our lab and evaluated the BBB permeability at 1×10^{10} vp/g. We found 0.34% transduction efficiency for AAV9 throughout the brain in the absence of insonation.

4) It is intriguing that AAV.FUS.3 may be a vector that is specifically optimized for FUS delivery. The authors suggest that it could be caused by either increased AAV.FUS.3 levels in circulation because of lower binding to other tissue; however, only the liver was investigated in peripheral organs.

4a) Please show evidence of transduction (or lack of transduction) in other peripheral organs, DRG, and/or the residual levels of AAV.FUS.3 in circulation.

In addition to the liver, we sectioned and imaged kidneys and lungs, and found no substantial transduction at these doses (in-text **lines 193-198**; Supplementary **Fig. S5**), which was consistent with the published reports for AAV9 (Kofoed et al, 2022).

Immunostaining of DRGs was not possible in our case – it led to high background that prevented quantitation when using tissue clearing and immunostaining, and sectioning was unsuccessful due to the small size of mouse DRGs (~ 1 mm tissue pieces after extraction).

5) The authors also suggest that AAV.FUS.3 may lead to sophisticated changes in receptor binding and BBB interaction facilitates the crossing of the BBB at FUS-targeted sites. This could represent a future study of interest. Another option, which the authors did not mention, is the possibility that AAV.FUS.3 is better than AAV9 in transducing neurons upon brain entry.

5a) This can easily be tested by comparing AAV9 and AAV.FUS.3 transduction following intracranial injections. Adding this data will significantly contribute to the understanding of whether AAV.FUS.3 is specifically improved for FUS-mediated delivery or improved for neuronal transduction overall.

Thank you for this interesting suggestion. To test this possibility, we performed intracranial/intraparenchymal injections to the hippocampus at a relevant dose and compared to FUS-BBBO-mediated delivery of both vectors. The improvement of neuronal transduction was comparable regardless of the route of delivery, suggesting that the AAV.FUS.3 is indeed better at transduction of brain cells overall. We thank the reviewer for their helpful suggestion which improved the manuscript.

The results are reported on **lines 262-266** and in **Supplementary Figure S10**.

6) The neuronal tropism of AAV.FUS.3 and its comparison with AAV9 represent several pitfalls that must be addressed.

6a) While AAV.FUS.3 may have enhanced neuronal tropism, AAV.FUS.3 transduced liver cells and expressed GFP. Additional data on the cellular tropism and expression of AAV.FUS.3 in peripheral organs, DRG, and in the brain (neuronal, glial and others) are important.

Within the brain, we counterstained for astrocytes/glia (GFAP) and microglia (Iba1) and AAV.FUS.3 relative to AAV9 (in-text **lines 217-221**; Supplementary **Fig. S7**). We saw comparable transduction of GFAP positive cells for both serotypes (8% vs 3.4% for AAV9 and AAV.FUS.3, $p=0.0552$; paired two-tailed t-test; $t=4.076$), and 5-fold reduction in transduction of Iba1-positive cells (3.5% vs 0.7% for AAV9 and AAV.FUS.3; $p=0.0174$; paired two-tailed t-test, $t=7.487$).

Outside of the brain, in addition to the liver, we sectioned and imaged kidneys and lungs, where no detectable transduction of both viral vectors was observed (**Supplementary Fig. S7**).

6b) Data should be presented and compared in absolute numbers for AAV9 and AAV.FUS.3 (and not percentage or fold-improvement). Since AAV9 and AAV.FUS.3 may transduce neurons and glia differently depending on the brain regions, their comparison for neuronal tropism (e.g. Fig 4) and across brain regions (e.g. Fig 5) as percentages and fold improvement is not always the most informative.

There is a number of reasons why we chose ratio of AAV9 and AAV.FUS as the outcome measure. First, this approach controls better for the BBB opening quality. For example, the number of transduced neurons depends on the quality of the BBB opening, tail-vein injection, and the vector itself. Since the AAV9 and AAV.FUS were co-injected after the FUS-BBBO, their ratio of transduction is only dependent on the vector properties, representing a more controlled way to evaluate the vectors, without the confounds of other variables. AAV9 is also the benchmark vector showing the best efficiency of transduction among naturally-occurring vectors (see e.g. Kofoed et al, 2022) which provides a point of reference to others in the field who are familiar with FUS-BBBO gene delivery. We clarified this on **lines 100-102**.

Nonetheless, we agree that absolute transduction values are also useful to other researchers in the field and have now provided them for each region in an additional panel in **Figure 5 g**. Additionally, we performed the same analysis for BALB/cJ mice in **Supplementary Figure S11e**. Finally, we also measured the overall transduction efficiency across the brain regions at a lower dose of AAV.FUS.3, in **Supplementary Figure S8d**. The improvement of neuronal transduction is consistent across the brain regions, suggesting that the tropism effect is not region-specific.

Examples (there are others) requiring improvements:

6b.1) The enhanced neuronal tropism of AAV.FUS.3 in 4c seems to be limited to a dense neuronal cell layer. Brain regions and subregions should be identified clearly and analyzed separately if heterogeneity is present. What are the other cell types transduced by AAV9 and AAV.FUS.3? Is this the hippocampal formation, and if yes were exactly?

This hippocampus image is just one of the examples which separates the non-neuronal cells spatially to make it easier for readers to see the difference. We provided a higher magnification image from the thalamus in **Fig. 4c**, and quantified the cell-tropism in other cell types as well in **Supplementary Figure S7**.

6b.2) In Figure 5, the important data to most readers may not be the fold improvement over AAV9 but the representative transduction efficiency for each vector, within each region. The raw

data would show this nicely instead of fold-improvement (fold-improvement is interesting and can be mentioned in the text instead). For example, looking at the fold-improvement data (5f), one may think that AAV.FUS.3 is not great for cortical transduction but, judging from 5a one can think that both AAV9 and AAV.FUS.3 seems to transduce cortical cells better than hippocampal cells. Co-localization with NeuN and other cell types would be very informative in Figure 5.

As suggested, we provided the neuronal transduction for each brain region in Fig. 5g. As described in the response to comment 6b, the fold-ratio of transduction compared to the benchmark vector is critical when evaluating the performance of the vector independent of other variables such as BBB opening efficiency, and we kept this data in as well.

7) The authors state on lines 50-51 that AAVs can be safely delivered to small and large areas using FUS. The authors should modify this statement, as this is only known for FUS treatment alone and not in combination with AAV. Though no gross tissue damage has been reported following FUS-mediated delivery of AAVs to the brain, most studies are not long-term, and especially the immune reactions that may arise following AAV delivery do not cause tissue damage within those timeframes.

We cited a paper published by our lab (Nouraein et al, 2023) on safe and successful gene delivery using FUS-BBBO to large brain volumes, and another one published by another group (Felix et al, 2021). We also clarified the damage was not detected only within the tested time frames. These changes are reported on **lines 55-56**.

8) It is unclear if Supp Fig S2 is in the brain or liver? Since tissue autofluorescence differs between organs and fluorophores, the reliability of the quantification method using GFP and mCherry should be included for both brain and liver.

We edited the manuscript to specify that this is in the brain on **line 173** and the figure caption.

9) Lines 241-244 the authors state that FUS loosens tight junctions and allows substances to pass from the blood to the brain. The authors should also mention other mechanisms, e.g., increased transcytosis and decreased levels of efflux transporters.

We edited the manuscript to include the stated mechanisms along with literature cited, on **lines 52-54**.

10) Please clarify the brain processing and analysis:

10a) Brains were sectioned at 50-micron but 100-micron sections were analyzed?

We corrected the text on **lines 484, 494** to indicate that 50-micron sections were analyzed.

10b) Which direction was chosen for brain sectioning? Horizontal, coronal or saggital?
Since the FUS spots only target some parts of the brain, the sectioning and sampling of sections for analysis will have a large impact on the data obtained.

We specified the coronal direction in text, on **line 484**.

10c) Which area was quantified? Were all positive cells in the entire section counted or only in the FUS targeted area?

We quantified positive cells only in the FUS-targeted area. The details regarding the selection of region of interest are provided in methods section **lines 497-500**.

11) Since evolution-based methods have previously shown the development of AAVs with species-specific properties, please include in the mouse description the strain of origin.

We specified the 3 mice strains that were used in this study in the animal description of the Materials and Methods section on **line 343**.

Thank you for your thorough and helpful review!

Best,

Jerzy Szablowski, Ph.D.
Assistant Professor,
Dept. of Bioengineering,
Rice Neuroengineering Initiative

Reviewers' Comments:

Reviewer #1:

Remarks to the Author:

The utilization of Focused Ultrasound (FUS) presents a promising method for achieving the opening of the Blood-Brain Barrier (BBB) to facilitate AAV delivery. This approach, coupled with Cre selection, allows for the engineering of AAV capsids to target the BBB opening while avoiding off-target effects in peripheral organs. This work has resulted in the development of novel engineered AAV capsids tailored for efficient gene delivery to the mouse brain, with a preference for neuronal targeting. According to reviewer's comments, author modified the context as well as the collected more data from another mouse strain.

Based on the revision, reviewer have following questions and comments.

1. Figure 2: Please confirm the number of the dark gray dots as "35."
2. Figure 3: While the engineered AAV.FUS demonstrated superior transduction compared to AAV9 (serving as an internal transgene expression control?), it is essential to emphasize the rationale behind this comparison via co-injection, considering potential disparities in the Gene of Interest (GOI). Additionally, differences in emission conditions for the two fluorescent markers (mCherry vs GFP) may impact the interpretation of the data collected. The reviewer suggests conducting long-term observations beyond the two-week window to further validate the observed superiority.
3. In addition to GFAP and IbaI, the reviewer strongly recommends the utilization of other cell markers to define the targeting neurons or localization.
4. Regarding the AAV injection route, this study demonstrated that the engineered capsids could accumulate at the BBB opening via intravenous injection without invasion, in contrast to other brain injection routes such as intraparenchymal injections. The reviewer seeks clarification on why intraparenchymal injections were chosen for the latter experiments (even could employ a lower dose in this way?).
5. Method on Focused Ultrasound Equipment and BBB Opening Procedures: For the library screening, what was the time interval between AAV injection and ultrasound?
6. Moreover, the reviewer proposes that alkaline gel titration could serve as a more robust method for titration, compared to the strategies described in lines 426-430.
7. Reviewer suggests that providing proof of the ratio between transgene expression and viral copy number among tissues would offer more informative data.
8. Please ensure consistent use of italics for "in vivo" in all contexts.
9. Please reconsider the use of " acoustically" in the title.
10. Did the author compare the Neutralizing Antibody (Nab) titer with engineered serotypes to other natural AAVs?
11. The AAV-transduced signals in Figure 3a reflects inconsistent with the data from Figure 5. The reviewer recommends including zoomed-out images for Figure 5 if possible.
12. Please specify whether transgene expression was determined by immunostaining or expressed fluorescence proteins.

Reviewer #2:

Remarks to the Author:

The authors have thoughtfully addressed the reviewers' comments, adding a remarkable amount of valuable data. It was a pleasure reviewing this improved version of the manuscript.

Points to consider.

1. The evaluation of AAVs at a lower dose is very informative. The discussion would benefit from briefly comparing the main results obtained at the two dosages.
2. The AAV dose units recommended are GC or VG; typically GC/kg
3. Please mention the doses of AAVs in all figure legends. This would be useful and clarify:
a. the 'various doses' mentioned in the legend Suppl Fig. 8c, and

b. the response to the reviewer "transduction at these doses" (lines 193-198; Supplementary Fig. S5)". Only one dose seems to be tested in Fig. S5.

4. Regarding peripheral transduction lines 312-313 and 327-335, please revise the sentence to include the DRG as part of the peripheral nervous system, in addition to peripheral organs.

5. Line 311 (or elsewhere), add DRG toxicity with appropriate references.

6. Please clarify in the manuscript that Iba1 is a microglia/macrophage marker. It is not specific to microglia.

7. If/where appropriate, the study by Blesa et al., *Sci. Adv.* 9, eadf4888 (2023) could be added

RICE UNIVERSITY
School of Engineering
Department of Bioengineering

Jerzy Szablowski, Ph.D.
Assistant Professor of Bioengineering
Rice Neuroengineering Initiative member
6566 Main St, Rice BRC, #869
Houston, TX 77030

Dear reviewers,

Please find the attached responses to your comments and questions. The responses are inset to improve readability. The changes to the manuscript are indicated in yellow in the main manuscript file.

Reviewer #1 (Remarks to the Author):

The utilization of Focused Ultrasound (FUS) presents a promising method for achieving the opening of the Blood-Brain Barrier (BBB) to facilitate AAV delivery. This approach, coupled with Cre selection, allows for the engineering of AAV capsids to target the BBB opening while avoiding off-target effects in peripheral organs. This work has resulted in the development of novel engineered AAV capsids tailored for efficient gene delivery to the mouse brain, with a preference for neuronal targeting. According to reviewer's comments, author modified the context as well as the collected more data from another mouse strain.

Based on the revision, reviewer have following questions and comments.

We thank the reviewer for helpful suggestions that have improved the manuscript, both during the last round of revision and the current one. Our answers to specific comments are inline.

1. Figure 2: Please confirm the number of the dark gray dots as "35."

The graph shows 22 points, the remaining 13 points do not appear on the graph because they showed zero transduction in untargeted hemispheres and cannot be shown on the log-log plot. The figure caption has been corrected. The enrichment after FUS-BBBO of these 13 clones was lower than any of the 22 clones that are displayed in the figure, and therefore they do not contain any of the top 5 clones selected for further validation.

The caption of **Figure 2** has been amended:

Dark grey dots represent 22 clones that are enriched in the FUS targeted hemispheres at least 100-fold in every tested mouse and DNA sequence encoding the 7-mer insertion peptide. Additional 13 clones had zero detected transduction in the untargeted hemisphere and could not be presented on the log-log plot.

2. Figure 3: While the engineered AAV.FUS demonstrated superior transduction compared to AAV9 (serving as an internal transgene expression control?), it is essential to emphasize the rationale behind this comparison via co-injection, considering potential disparities in the Gene of Interest (GOI). Additionally, differences in emission conditions for the two fluorescent markers (mCherry vs GFP) may impact the interpretation of the data collected. The reviewer suggests conducting long-term observations beyond the two-week window to further validate the observed superiority.

Thank you for the suggestion. We believe that cell counting in animals co-transduced with two vectors provides the most reliable basis for quantitative comparison. This counting relies on detection of positive cells rather than measurement of fluorescence levels, which is more robust to changes in expression over time. To ensure that such counting is reliable, we performed co-transduction with AAV9-mCherry and AAV9-GFP and showed that cell counts were consistent between the two GOIs. This data is shown in Supplementary Figure S2.

By employing co-injection, we were able to ensure that the transduction efficiency being measured is due to qualities of the vector, not the variables related to the BBB opening, or intravenous injections. First, Co-injection controls for efficiency of FUS-BBBO. If BBB opening is weaker in an animal, both the AAV9 and AAV.FUS fluorescence will be lower. If each mouse received just one vector at a time, there would be no way to distinguish poor BBB opening from lower efficiency of transduction. Second, we initially hypothesized that the transduction efficiency could be dependent on the brain region. Since FUS-BBBO location can vary slightly (~1 mm) from mouse to mouse, we wanted to ensure the AAV9 and the AAV.FUS candidates are delivered to precisely the same brain region to allow for a fair comparison.

We have updated the manuscript **on line 168** to clarify this rationale.

3. In addition to GFAP and Iba1, the reviewer strongly recommends the utilization of other cell markers to define the targeting neurons or localization.

We agree that additional staining would be helpful and added a stain for oligodendrocytes, which showed significantly lower transduction with AAV.FUS.3 compared to AAV9 (21.9-fold, $p < 0.0001$, two-tailed paired t-test, $n = 6$ mice). While future studies can stain or sequence for additional cell sub-types, we believe our stains now cover the most prevalent neuronal and glial cell types in the mammalian brain. Of course, in the future characterizing the tropism of specific neuronal subtypes would warrant further study.

The new data is presented in Supplementary Figure S7c, and referred to **on lines 223-224**.

4. Regarding the AAV injection route, this study demonstrated that the engineered capsids could accumulate at the BBB opening via intravenous injection without invasion, in contrast to other brain injection routes such as intraparenchymal injections. The reviewer seeks clarification on why intraparenchymal injections were chosen for the latter experiments (even could employ a lower dose in this way?).

The comparison of our method to intraparenchymal injection was requested by one of the reviewers during the last round of revisions. The goal was to identify whether AAV.FUS' improved transduction in the brain could be due to its improved efficiency of neuronal transduction once the vector is in the brain. As a side-benefit, the increased efficiency of AAV.FUS compared to AAV9 in intraparenchymal transduction could be useful to researchers using this delivery route rather than FUS-BBBO. We have updated the text **on line 266-269** to clarify these points.

5. Method on Focused Ultrasound Equipment and BBB Opening Procedures: For the library screening, what was the time interval between AAV injection and ultrasound?

Thank you for mentioning this point, as it is important to clarify for the readers. We clarified it in Methods and indicated that the vector injection was done within 2 minutes of the BBB opening, which was the time needed to mix the microbubbles and inject them into the previously-placed tail vein catheter.

The new methods section now states:

Subsequently, the mice were injected via tail vein with AAVs. Within two minutes after viral injection, the mice were also injected with 1.5×10^6 DEFINITY microbubbles (Lantheus) and 0.125 μmol of ProHance (Bracco Imaging) dissolved in sterile saline, per g of body weight.

The change is **on line 375**.

6. Moreover, the reviewer proposes that alkaline gel titration could serve as a more robust method for titration, compared to the strategies described in lines 426-430.

Thank you for this suggestion. We agree that additional titering measures could be helpful beyond the qPCR used in our study. However, we think it is important to point out that our study design controlled for any potential variability in titers. First, we only co-injected AAVs packaged by the same manufacturer at the same time using the most commonly used methodology that will likely be used by many other ultrasound labs. We continued this strategy throughout the study to ensure the consistency of titration methodology. Second, we used the ratio of brain and the liver transduction as the primary measure of efficacy because it is dependent on the viral vector tropism properties, rather than the vector titer. For example, AAV.FUS.3 shows higher transduction in the brain, but lower in the liver, which cannot be explained by e.g. lower titer of AAV9 alone.

We agree that in future studies, additional titering measures such as alkaline gel titration will be helpful. Having used up the viral particles used in this manuscript, we cannot perform additional titering analyses. However, for the reasons stated above, we are confident that the titering and controls we performed are adequate to robustly support the conclusions of our study.

The methods were amended to provide further clarification **on lines 444-446**.

7. Reviewer suggests that providing proof of the ratio between transgene expression and viral copy number among tissues would offer more informative data.

We thank the reviewer for this suggestion. We elected to focus on counting of transduced cells because it allowed to focus our counting in the brain on regions that were exposed to FUS-BBBO (or contralateral controls) and to quantify the cell types that were transduced through correlative staining. Performing genome copy number measurements would introduce confounds due the need to accurately dissect targeted areas and would provide no information about cell type tropism. However, we agree that in future studies, especially in larger animals, additional methods to quantify transduction would be valuable.

8. Please ensure consistent use of italics for "in vivo" in all contexts.

Confirmed, all "in vivo" mentions are italicized. We thank the reviewer for noticing this mistake and appreciate the thorough review.

9. Please reconsider the use of " acoustically" in the title.

We thank the reviewer for the suggestion. We will consult with the editor regarding the title.

10. Did the author compare the Neutralizing Antibody (Nab) titer with engineered serotypes to other natural AAVs?

We did not, as we used naive mice that were not exposed to AAVs throughout the study, since such exposure would drastically reduce the brain uptake in mice, as seen in one of our previous studies (1).

References:

- 1) Chen, Maria, et al. "Immune profiling of adeno-associated virus response identifies B cell-specific targets that enable vector re-administration in mice." *Gene therapy* 30.5 (2023): 429-442.

11. The AAV-transduced signals in Figure 3a reflects inconsistent with the data from Figure 5. The reviewer recommends including zoomed-out images for Figure 5 if possible.

We provided zoomed-out images in the Supplementary Figures S10 (C57Bl6j mice) and S13 (Balb/c mice).

12. Please specify whether transgene expression was determined by immunostaining or expressed fluorescence proteins.

We thank the reviewer for suggesting an important clarification. The transgene expression was determined by expressed fluorescence of the proteins. This has been clarified in the methods, section **on line 514**.

Reviewer #2 (Remarks to the Author):

The authors have thoughtfully addressed the reviewers' comments, adding a remarkable amount of valuable data. It was a pleasure reviewing this improved version of the manuscript.

Points to consider.

We thank the reviewer for useful comments throughout this process. It made the manuscript substantially stronger and we are grateful for the help.

1. The evaluation of AAVs at a lower dose is very informative. The discussion would benefit from briefly comparing the main results obtained at the two dosages.

We thank the reviewer for this suggestion. We now expanded the discussion to talk about the scaling of transduction with dose, referencing previous literature.

The discussion has been amended **on lines 319 to 324**.

2. The AAV dose units recommended are GC or VG; typically GC/kg

We have corrected this throughout the manuscript.

3. Please mention the doses of AAVs in all figure legends. This would be useful and clarify:

a. the 'various doses' mentioned in the legend Suppl Fig. 8c, and

We have clarified that this was one dose in this figure.

b. the response to the reviewer "transduction at these doses" (lines 193-198; Supplementary Fig. S5)". Only one dose seems to be tested in Fig. S5.

That is of course correct, and the caption has been fixed.

4. Regarding peripheral transduction lines 312-313 and 327-335, please revise the sentence to include the DRG as part of the peripheral nervous system, in addition to peripheral organs.

Thank you! We have corrected the text as follows.

(Lines 325-326)

Larger-scale studies will be needed to evaluate peripheral transduction and toxicity in all peripheral organs and peripheral nervous system such as dorsal root ganglia (DRG)^{38, 58}, in large animal species before potential translation of AAV.FUS.3.

(Lines 347-348)

The current limitations of gene therapies include: commonality of pre-existing neutralizing antibodies in a large fraction of the population⁶⁶; high liver transduction leading to toxicity⁴³ and potential carcinogenicity⁶⁷; AAV-induced toxicity in DRGs in primates^{38, 58}; and high cost of the therapy⁶⁸.

5. Line 311 (or elsewhere), add DRG toxicity with appropriate references.

Added two references **on line 348**.

6. Please clarify in the manuscript that Iba1 is a microglia/macrophage marker. It is not specific to microglia.

Clarified

7. If/where appropriate, the study by Blesa et al., Sci. Adv. 9, eadf4888 (2023) could be added

We added the citation of this landmark study showing AAV delivery to the brains of non-human primates (NHPs) in the discussion as it is highly relevant to the future of FUS-BBBO gene delivery.

It is now added **on lines 343-345** as:

Recently, a groundbreaking study has also shown that delivery of AAV9 into the brain can be achieved in non-human primates using FUS-BBBO⁶⁵, further bolstering the translational potential of this procedure.

Best,

Jerzy Szablowski, Ph.D.
Assistant Professor,
Dept. of Bioengineering,
Rice Neuroengineering Initiative

Reviewers' Comments:

Reviewer #1:

Remarks to the Author:

Authors addressed most of reviewers's comments, however, review need some clarification on below concerns.

1. Fig S2 & line 168: Author applied AAV9/mChery as a housekeeper for evaluation on the AAV.Fus/GFP' transduction and tropism. Here, is this data from single vector's injection with different doses of each vector? It's better to show the method or detail description on this data collecting.

2. Fig4: To emphasis the specific neuronal tropism of AAV.Fus, reviewer suggested to use the venn diagram to show the overlap between AAV9 and AAV.Fus.3, as well as the overlap between neuN+ and GFP+ or mCherry+.

Reviewer #2:

Remarks to the Author:

The authors have addressed all of the reviewer's comments. Thank you.

Dear reviewers,

We appreciate your thorough review and the comments which substantially improved the manuscript. Please find the attached responses to your comments and questions. The responses are inset to improve readability. The changes to the manuscript are indicated in yellow in the main manuscript file.

Authors addressed most of reviewers's comments, however, review need some clarification on below concerns.

1. Fig S2 & line 168: Author applied AAV9/mChery as a housekeeper for evaluation on the AAV.Fus/GFP' transduction and tropism. Here, is this data from single vector's injection with different doses of each vector? It's better to show the method or detail description on this data collecting.

This is the data from injection of vectors injected at the same dose, variability happened because of the difference in transduction efficiency across different mice and brain regions. We provided the dose data and further clarification **on lines 167-168** and in the caption of Figure S2.

2. Fig4: To emphasis the specific neuronal tropism of AAV.Fus, reviewer suggested to use the venn diagram to show the overlap between AAV9 and AAV.Fus.3, as well as the overlap between neuN+ and GFP+ or mCherry+.

We thank the reviewer for this suggestion. We reviewed how typically this type of data is presented and found the scatter plots and bar graphs are typical for the field of vector engineering to show relative transduction efficiency (e.g. see refs 1-3 below). This method shows individual mice and thus can inform the reader about the variability in the neuronal transduction specificity.

References:

1. Deverman, Benjamin E., et al. "Cre-dependent selection yields AAV variants for widespread gene transfer to the adult brain." *Nature biotechnology* 34.2 (2016): 204-209.
2. Li, Wuping, et al. "Engineering and selection of shuffled AAV genomes: a new strategy for producing targeted biological nanoparticles." *Molecular Therapy* 16.7 (2008): 1252-1260.
3. Zhu, Danqing, et al. "Optimal trade-off control in machine learning-based library design, with application to adeno-associated virus (AAV) for gene therapy." *Science Advances* 10.4 (2024): eadj3786.

Reviewer #2 (Remarks to the Author):

The authors have addressed all of the reviewer's comments. Thank you.

We thank the reviewer for the helpful suggestions throughout this process!